# Microstructural and crystallographic evolution of palaeognath (Aves) eggshells

Seung Choi[1,2]*, Mark E Hauber[3], Lucas J Legendre[4], Noe-Heon Kim[5,6], Yuong-Nam Lee[5], David J Varricchio[1]

[1]Department of Earth Sciences, Montana State University, Bozeman, United States; [2]Key Laboratory of Vertebrate Evolution and Human Origins of Chinese Academy of Sciences, Institute of Vertebrate Paleontology and Paleoanthropology, Chinese Academy of Sciences, Beijing, China; [3]Department of Evolution, Ecology, and Behavior, School of Integrative Biology, University of Illinois Urbana-Champaign, Urbana, United States; [4]Department of Geological Sciences, University of Texas at Austin, Austin, United States; [5]School of Earth and Environmental Sciences, Seoul National University, Seoul, Republic of Korea; [6]Department of Geosciences, Princeton University, Princeton, United States

**Abstract** The avian palaeognath phylogeny has been recently revised significantly due to the advancement of genome-wide comparative analyses and provides the opportunity to trace the evolution of the microstructure and crystallography of modern dinosaur eggshells. Here, eggshells of all major clades of Palaeognathae (including extinct taxa) and selected eggshells of Neognathae and non-avian dinosaurs are analysed with electron backscatter diffraction. Our results show the detailed microstructures and crystallographies of (previously) loosely categorized ostrich-, rhea-, and tinamou-style morphotypes of palaeognath eggshells. All rhea-style eggshell appears homologous, while respective ostrich-style and tinamou-style morphotypes are best interpreted as homoplastic morphologies (independently acquired). Ancestral state reconstruction and parsimony analysis additionally show that rhea-style eggshell represents the ancestral state of palaeognath eggshells both in microstructure and crystallography. The ornithological and palaeontological implications of the current study are not only helpful for the understanding of evolution of modern and extinct dinosaur eggshells, but also aid other disciplines where palaeognath eggshells provide useful archive for comparative contrasts (e.g. palaeoenvironmental reconstructions, geochronology, and zooarchaeology).

*For correspondence:
seung0521@gmail.com

## Editor's evaluation

This fundamental study represents a significant advance in our understanding of the complex evolutionary history of the eggshell features in one of the main living bird lineages, Palaeognathae, with compelling and thoughtfully presented results. The work will be of interest to many biologists, paleontologists, and archaeologists.

## Introduction

Non-avian dinosaurs became extinct at the end of Mesozoic (*During et al., 2022*), but avian dinosaurs are still extant and flourishing today as the most speciose land-vertebrate lineage (*Jarvis et al., 2014*; *Lee et al., 2014*; *Brusatte et al., 2015*; *Prum et al., 2015*). For example, their eggshell sizes, shapes, and pigment colours and patterns show phenomenal diversity (*Mikhailov, 1997a*; *Hauber et al., 2014*; *Stoddard et al., 2017*). In the palaeontological record, even more diverse forms of

**eLife digest** About 50 species of birds on the planet today do not belong to the same group as the other 10,000 currently in existence. Known as the paleognaths, this small clade features many of the largest and heaviest avian specimens on Earth, bringing together ostriches and their distant South American relatives the rheas, as well as emus and cassowaries. Kiwis and ground-dwelling species known as tinamous complete the family. None of these birds can fly, except for the tinamous. Paleognath eggs are also somewhat distinct from the rest of the avian population, being larger and sporting thicker shells. Advanced genetic analyses in the late 2000's have upended researchers' understanding of in what sequence these birds have evolved, and how they are related to each other. The new phylogenetic family tree offers the opportunity to re-evaluate previous conclusions about this group, which could in turn clarify the evolution and lifestyle of flightless modern and extinct dinosaurs.

Choi et al. decided to use this updated genetic information to better understand how paleognath eggs have evolved. Traditionally, these have been loosely classified into three types (rhea-style, ostrich-style and tinamou-style) based on various morphological features. Their microstructure, however, remains poorly studied, and it is unclear whether this categorisation reflects evolutionary processes. Aiming to fill this gap, Choi et al. employed electron microscopy approaches to examine the microstructure of the eggshell in all groups of paleognath birds (including the now extinct moas from New Zealand and elephant birds from Madagascar), as well as in selected species of flying birds and non-avian dinosaurs. Combined with the new evolutionary tree and additional analyses, these experiments suggest that the ancestor of the paleognaths laid rhea-style eggs, which are still the most common type amongst the family. In fact, several non-paleognath bird eggs also showed these features. In contrast, ostrich-style and tinamou-style eggs seem to have evolved independently in several distantly related species within the group. Equipped with this knowledge, it may become possible for ornithologists to decipher how eggshells evolved in other lineages of flightless birds, and for palaeontologists to better interpret fossil bird and other dinosaur eggs.

dinosaur eggs and eggshells have been recovered (*Mikhailov, 1997b*; *Grellet-Tinner et al., 2006*; *Norell et al., 2020*; *Oser et al., 2021*), with many divergent designs disappearing with the extinction of non-avian dinosaurs. Eggshell is mainly a biomineral (calcium carbonate; $CaCO_3$ with an inner proteinaceous shell membrane and outer cuticle cover; *Kulshreshtha et al., 2022*) that can be well-preserved in the fossil record (*Carpenter and Alf, 1994*; *Mikhailov and Zelenkov, 2020*). Thereby, dinosaur eggs are a valuable subject for evolutionary biology for the tracking of phenotypic changes over geological timescales from the Early Jurassic onward (*Stein et al., 2019*; *Choi and Lee, 2019*). However, how eggs and eggshells evolve in a single dinosaur clade is not well understood and homoplastic similarities can obscure understanding. Documenting the evolution of eggs among modern avian dinosaurs provides helpful insights into the evolution among extinct taxa.

Palaeognathae is one of two major clades of modern birds (or modern dinosaurs) (*Jarvis et al., 2014*). Extant Palaeognathae are usually larger than their sister-clade Neognathae and are flightless except for the poorly flighted tinamous (*Yonezawa et al., 2017*; *Altimiras et al., 2017*). However, the Palaeogene palaeognaths Lithornithidae might have been a fully volant clade (*Torres et al., 2020*; *Widrig and Field, 2022*). Proportional to their body size, absolute sizes of eggs and eggshells of Palaeognathae are usually large and thick, respectively (*Grellet-Tinner, 2006*; *Birchard and Deeming, 2009*; *Legendre and Clarke, 2021*). Furthermore, eggshells of Palaeognathae show distinctive microstructures compared to those of Neognathae (*Mikhailov, 1997a*; *Zelenitsky and Modesto, 2003*; *Grellet-Tinner, 2006*). Species diversity of Palaeognathae is much lower than that of Neognathae (*Prum et al., 2015*), but it is critical for avian egg research. Due to this low number of species, it is feasible to investigate the egg features of all major clades of Palaeognathae. More importantly, considering that most Palaeognathae are flightless and that flight influences egg mass in Dinosauria (*Legendre and Clarke, 2021*), which influences eggshell thickness (*Ar et al., 1979*; *Legendre and Clarke, 2021*), the eggs and eggshells of Palaeognathae might be more appropriate modern analogues for those of flightless non-avian dinosaurs than those of volant Neognathae.

Previous studies of palaeognath eggs and eggshells (*Zelenitsky and Modesto, 2003*; *Grellet-Tinner, 2006*) interpreted the features in the light of morphology-based phylogenies of Palaeognathae.

However, the phylogeny of Palaeognathae has drastically changed since the late 2000s, mainly due to the advancements of molecular approaches (*Harshman et al., 2008*; *Phillips et al., 2010*; *Mitchell et al., 2014*; *Grealy et al., 2017*; *Yonezawa et al., 2017*; *Sackton et al., 2019*; *Cloutier et al., 2019*). For example, morphological phylogenies interpreted tinamous as the sister clade of all other Palae-ognathae (*Livezey and Zusi, 2007*), but a more recent molecular view regards tinamou and moa as sister clades (*Phillips et al., 2010*), which are a less inclusive clade within Palaeognathae (*Sackton et al., 2019*). Thus, revisiting eggs and eggshells of Palaeognathae with a revised phylogeny of Palae-ognathae is a timely issue for a comprehensive and updated understanding of avian egg evolution.

Tracing the evolution of palaeognath eggs and eggshells is important for both ornithology and palaeontology. For ornithology, it provides a representative case on how the macro-, microstructure and crystallography of eggs have evolved in the avian clade where speciation timelines are now avail-able owing to molecular clocks (*Yonezawa et al., 2017*). Based on a recent evolutionary scenario, stating that diverse palaeognath clades lost flight and acquired gigantism independently (*Harshman et al., 2008*; *Phillips et al., 2010*; *Mitchell et al., 2014*; *Grealy et al., 2017*; *Yonezawa et al., 2017*; *Sackton et al., 2019*; *contra Cracraft, 1974*), palaeognath eggshells provide chances to appraise potential homologies (similarities inherited from the most recent common ancestor [MRCA]) and homoplasies (similarities caused by similar selective regime [or neutral factors] rather than common ancestry; see *Patterson, 1988*; *de Pinna, 1991*; *Losos, 2011* for further information) in the evolution of avian eggshells.

For palaeontology, firstly, palaeognath eggs offer the chance to track appearance/disappearance and character change rates of phenotypes in modern dinosaur eggs. The molecular-clock-based approach is impossible for non-avian dinosaur eggs because DNA is not preserved in Mesozoic fossils (*Saitta et al., 2019*) or at least non sequenceable (*Bailleul and Li, 2021*). Consequently, in palaeon-tology, inferring the evolutionary pathways of eggshell must rely on a phylogeny of egg-layers that is solely based on morphological traits, but morphology-based trees can conflict with molecular data (e.g. *Bunce et al., 2009*; *Darlim et al., 2022*; see also *Wiens, 2004*; *Quental and Marshall, 2010*; *Lee and Palci, 2015*). Moreover, morphological phylogenetic trees generated in palaeontology are usually based on insufficient chronologies due to the shortage of absolute radiometric age data. In this sense, palaeognath eggs and eggshells can be helpful modern exemplars for palaeontology that show how, and at what rate, evolution has worked in flightless dinosaur eggs.

Secondly, in the Cenozoic deposits, palaeognath eggshells have been frequently reported in diverse regions of the world (*Sauer, 1972*; *Harrison and Msuya, 2005*; *Bibi et al., 2006*; *Worthy et al., 2007*; *Patnaik et al., 2009*; *Donaire and López-Martínez, 2009*; *Wang et al., 2011*; *Pickford, 2014*; *Blinkhorn et al., 2015*; *Mikhailov and Zelenkov, 2020*). However, most previous investiga-tions focused on the thickness and pore canal structures on the outer surface of eggshells. They are useful information, but a few published images show peculiar microstructure (e.g. *Donaire and López-Martínez, 2009*; *Patnaik et al., 2009*; *Wang et al., 2011*), which are not observed in modern palaeognath eggshells (see below). It means that there may be diverse eggshell microstructures in the fossil record and to understand the difference correctly, a more comprehensive, baseline under-standing of the microstructure and crystallography of modern palaeognath eggshells is necessary.

Here we: (i) document the microstructure and crystallography of all major clades of palaeognath eggshells (and several selected neognath and non-avian dinosaur eggshells for comparison) using three different mapping techniques acquired by electron backscatter diffraction (EBSD), a state-of-the-art tool for eggshell microstructural and crystallographic study; (ii) reinterpret the evolution of palaeognath eggshells based on the radically revised phylogeny of Palaeognathae; (iii) discuss the implications of evolution of palaeognath eggshells for the aforementioned research areas; and (iv) suggest future research topics for which this study can be a basis.

## Results
### Description for EBSD maps
#### Ostrich (*Struthio camelus*)
The overall microstructure and crystallography are peculiar compared to other palaeognath eggshells (*Figure 1*; *Zelenitsky and Modesto, 2003*; *Grellet-Tinner, 2006*). The entire thickness of the eggshell is composed of prismatic calcite grains (*Zelenitsky and Modesto, 2003*; *Choi et al., 2019*). The

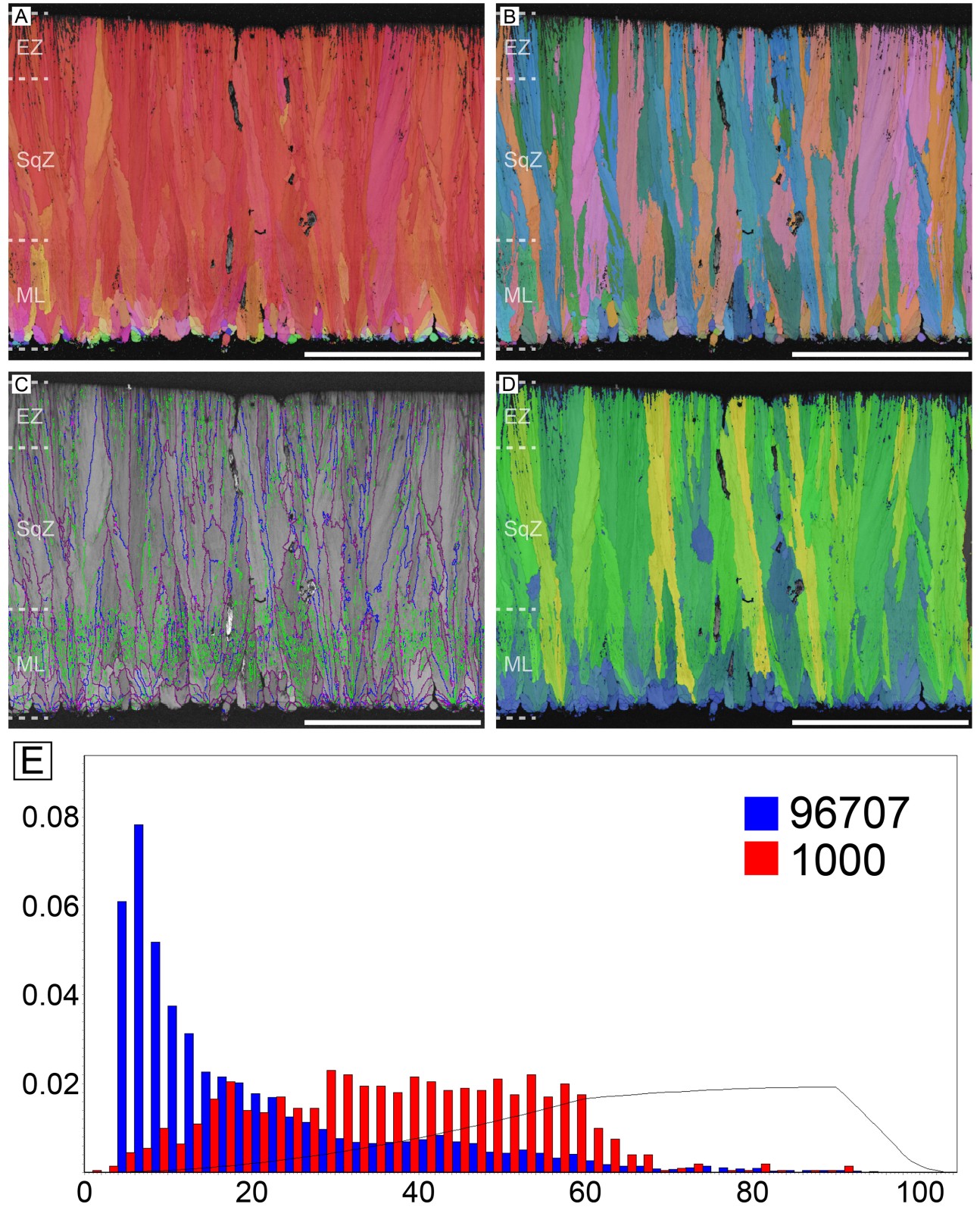

**Figure 1.** Ostrich eggshell. (**A**) IPF, (**B**) Euler, (**C**) GB, and (**D**) AR mappings (see *Figures 13–15* for legends). The dashed lines in the maps mark the boundary between the ML and SqZ; SqZ and EZ. Scale bars equal 1000 µm. (**E**) A misorientation histogram. The numbers in x- and y-axis represent degree between the two selected grains (either adjacent [blue; neighbour-pair method] or random [red; random-pair method]; see also *Figure 16*) and

*Figure 1 continued on next page*

*Figure 1 continued*

frequency, respectively. The numbers at the upper right corner mean the number of selected grains in each selection method. The explanation herein is applicable to the *Figures 2–12*, *Appendix 3—figures 1–5*, and *Appendix 4—figures 1–4*.

The online version of this article includes the following figure supplement(s) for figure 1:

**Figure supplement 1.** Composite GB mappings of fossil and modern eggshells that have prismatic shell unit structure.

mammillary layer (ML) is composed of wedge-like calcite grains and these grains usually extend to the outer edge of the eggshell. Low-angle (<10°; green lines in *Figure 1C*) grain boundaries (GB) are widespread in the eggshell, but they are concentrated at the outer part of ML (the existence and portion of ML is clearer in polarized light microscopic and scanning electron microscopic images; *Dauphin et al., 2006*). This feature has not been reported in any other avian and non-avian maniraptoran eggshell. Unlike most other avian eggshells that have rugged GB in squamatic zone (SqZ) and linear GB in external zone (EZ) (*Grellet-Tinner et al., 2012*; *Grellet-Tinner et al., 2016*; *Grellet-Tinner et al., 2017*; *Choi et al., 2019*), the GB in the SqZ of ostrich eggshell are seemingly linear. This makes it hard to identify the boundaries between the ML and SqZ, and between the SqZ and EZ in inverse pole figure (IPF) Y and Euler maps (*Zelenitsky and Modesto, 2003*; *Mikhailov, 2014*). However, EBSD provides highly-magnified images such that weakly developed rugged GB in the SqZ and slightly more linear GB in the EZ are observed (*Figure 1—figure supplement 1*). Thus, we support the view that there is a SqZ/EZ boundary near the outer surface of eggshell (*Mikhailov, 2014*). The peculiar prismatic microstructure of ostrich eggshell might have been derived from weakened development of squamatic ultrastructure and 'splaying' calcite growth.

## Rhea (*Rhea* sp.)

Rhea possesses microstructure and crystallography common to most palaeognath eggshells (*Figure 2*). The ML is comparatively thick and composed of wedge-like calcite. The boundary between the ML and SqZ is clear and can be identified by the contrasting microstructures. The SqZ is characterized by 'splaying' of the grain shape. There is crystallographic continuity between the SqZ and EZ, but calcite crystals in the EZ are usually prismatic in shape. Note that the overall microstructure is nearly the same as that of *Sankofa pyrenaica* and *Pseudogeckoolithus*, which are Late Cretaceous ootaxa (fossil eggtypes) from Europe (*López-Martínez and Vicens, 2012*; *Choi et al., 2020*). Low-angle GB are mostly concentrated in the SqZ. In ML and EZ, GB are linear, while in SqZ, GB are highly rugged. This trait can be observed even in simple secondary electron SEM images of rhea eggshells (*Choi et al., 2019*).

## Emu (*Dromaius novaehollandiae*) and cassowary (*Casuarius casuarius*)

Microstructure and crystallography of both genera are nearly the same, thus, they are described together (*Figures 3 and 4*). The major differences are that cassowary eggshell has a higher density of calcite grains in their ML and SqZ and presence of EZ is less clear. The ML of both genera has wedge-like calcite. The boundary between the ML and SqZ is clear. SqZ is characterized by 'splaying' of the grain shape. The SqZ is changed into EZ in outer region of compact part of the eggshell, which is characterized by different GB conditions. The 'resistant zone' (sensu *Zelenitsky and Modesto, 2003*; see *Figure 3—figure supplement 1*) of the eggshell shows crystallographic continuity with EZ. Therefore, we suggest that the 'resistant zone' be interpreted as a modified EZ that acquired porosity (*Figure 3—figure supplement 1*). This view is different from that of *Zelenitsky and Modesto, 2003* who regarded 'resistant zone' as a modified SqZ. Additionally, *Grellet-Tinner, 2006* interpreted this 'resistant zone' as a 'third layer' (=EZ in our terminology), which is partly in agreement with our view. In contrast, we suggest that the outer part of the 'second layer' (=SqZ in our term) in *Grellet-Tinner, 2006* is, in fact, part of EZ. Another unique feature of cassowary and emu eggshells is their 'granular layer' (sensu *Mikhailov, 1997a*; *Figure 3—figure supplement 1*). Although this layer was consistently reported in earlier studies (*Mikhailov, 1997a*; *Zelenitsky and Modesto, 2003*; *Grellet-Tinner, 2006*), it was usually treated as a layer simply overlying the porous EZ. However, this layer has a deep triangular 'root' to the middle of eggshell (*Lawver and Boyd, 2018*; *Choi et al., 2020*; note that the granular layer begins in SqZ in cassowary eggshell but in EZ in emu eggshell in our *Figures 3 and 4*, but it was not clade-specific and variable in both eggshells). Note that except for 'resistant zone' and granular layer, the overall microstructure of both emu and cassowary eggshells is very similar to

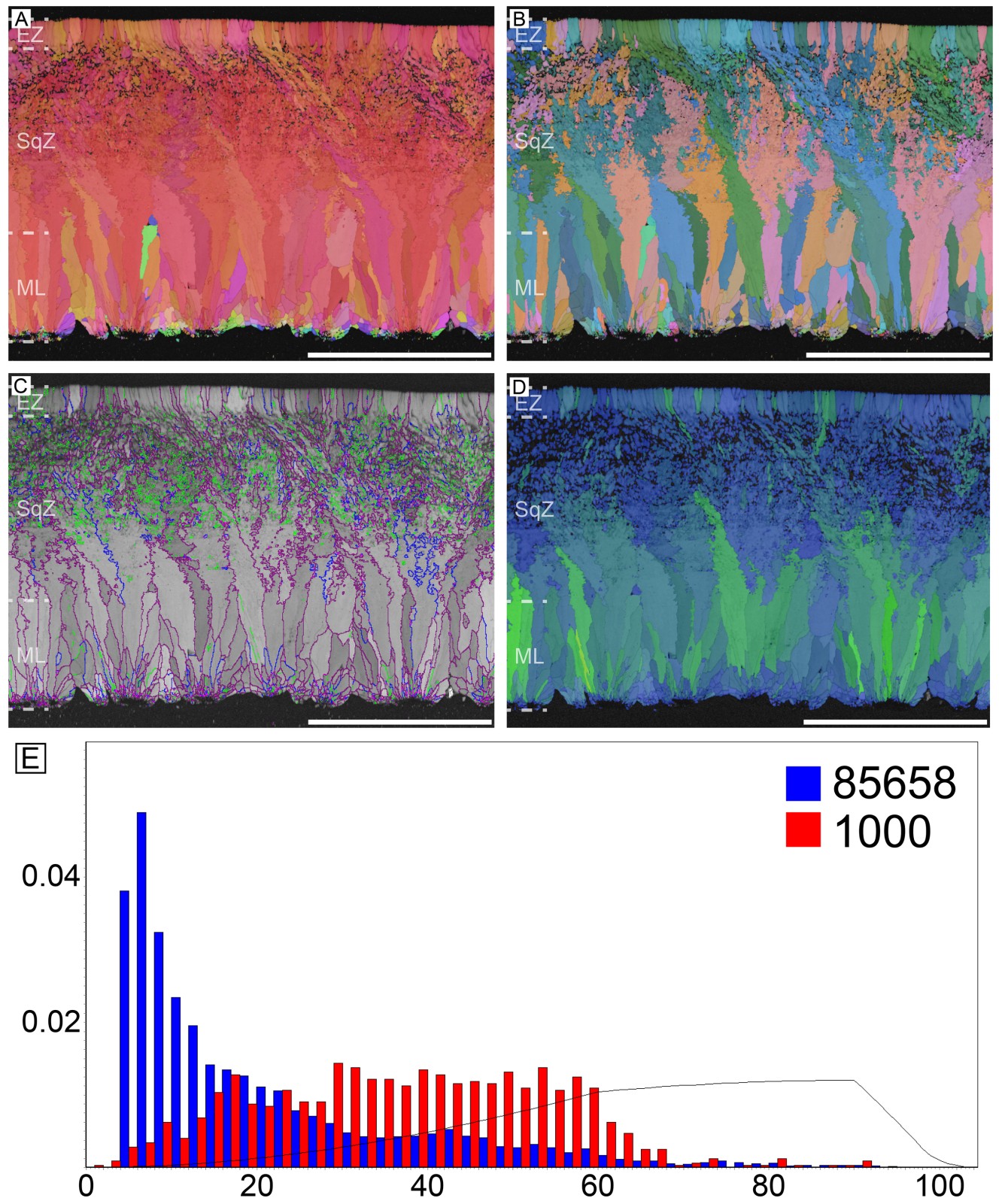

**Figure 2.** Rhea eggshell. Scale bars equal 500 μm.

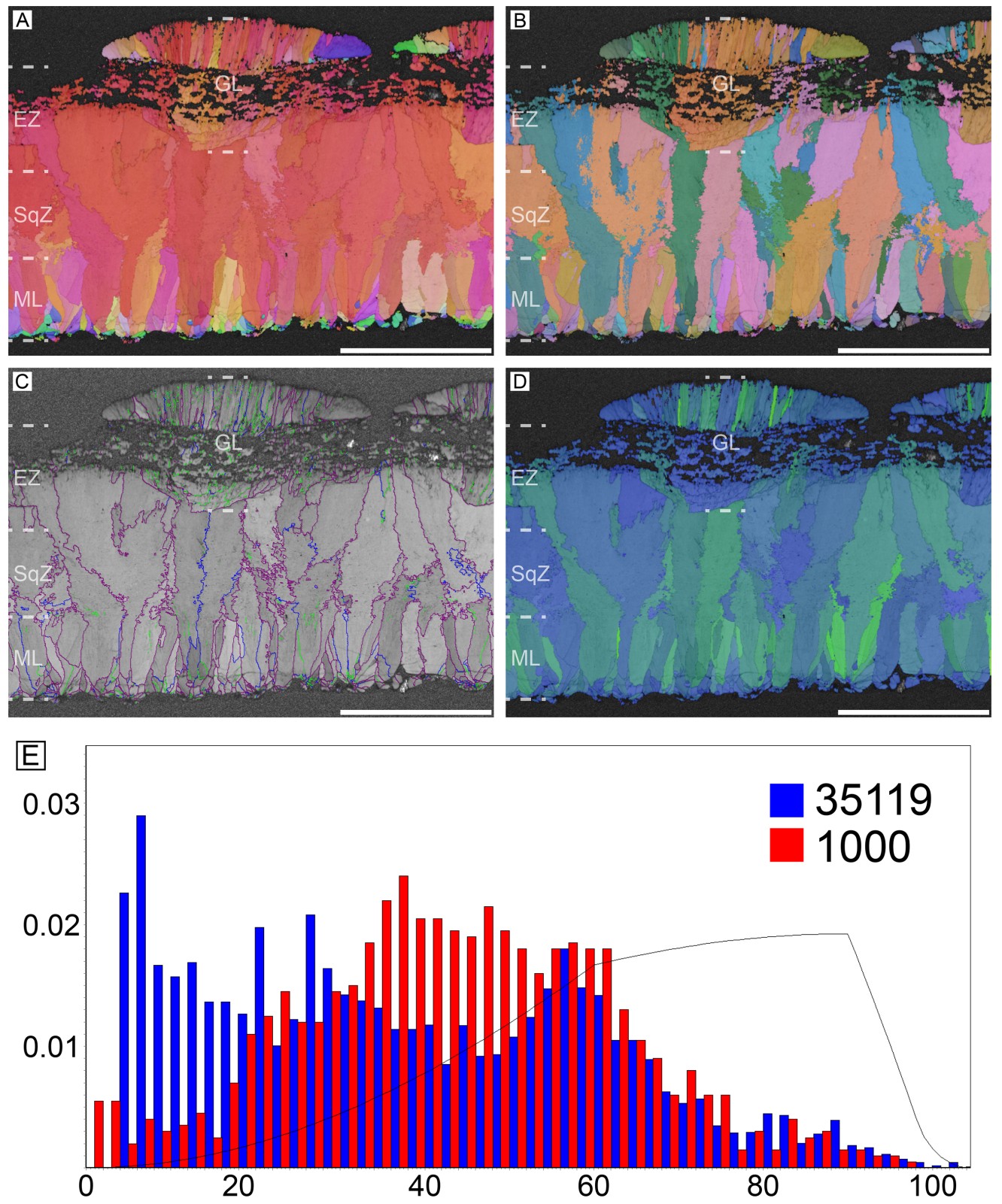

**Figure 3.** Emu eggshell. Note that deposition of granular layer (GL) begins in the EZ. Scale bars equal 500 µm.

The online version of this article includes the following figure supplement(s) for figure 3:

**Figure supplement 1.** Previous and current interpretations on the microstructure of casuariid eggshells.

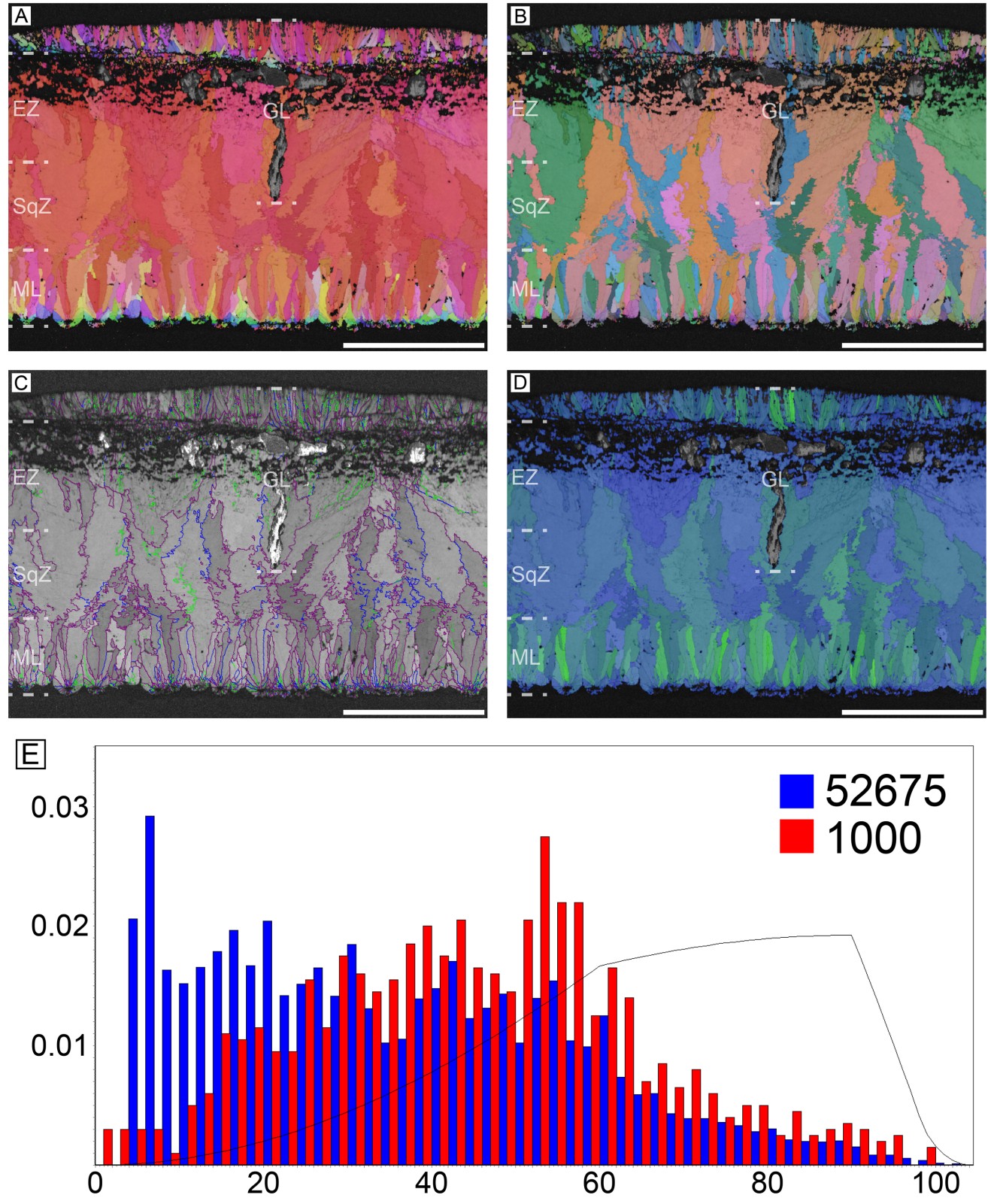

**Figure 4.** Cassowary eggshell. Note that deposition of granular layer (GL) begins in the middle of SqZ. Scale bars equal 500 μm.

those of *Pseudogeckoolithus* and *Sankofa* (*López-Martínez and Vicens, 2012*; *Choi et al., 2020*). Dissimilar to the rhea eggshell, the low-angle GB are not concentrated in SqZ, but usually present in granular layer (both outer granular part and its 'root'). In ML, GB are linear. The GB becomes highly rugged in the SqZ. The GB becomes linear again before they reach 'resistant zone', but this pattern is more prominent in emu eggshells. In the outer granular layer, GB are usually linear and lie parallel to each other.

### Kiwi (*Apteryx mantelli*)

In ML, grains are mostly wedge-like but in some parts, the width of the grains is very narrow so that needle-like (acicular) in shape (*Figure 5*; *Zelenitsky and Modesto, 2003*; but see *Grellet-Tinner, 2006*). The contour of ML is usually round. The boundary between the ML and SqZ is clear due to the contrasting grain shapes but the boundary can be extended into the middle of eggshell. The EZ crystals are massive and comparatively thick. Low-angle GB are mostly situated at the ML but low-angle GB are not abundant unlike other palaeognath eggshells. In ML and EZ, GB are linear. In SqZ, the GB are highly rugged.

### Elephant bird (Aepyornithidae)

As the largest known avian egg, it has the thickest eggshell (3.8 mm in average) (*Figure 6*; *Schönwetter, 1960*, *Ar et al., 1979*; *Juang et al., 2017*). Its microstructure and crystallography are closely similar to those of rhea eggshell despite the difference in thickness (*Figure 2*). The ML is composed of wedge-like calcite. The boundary between the ML and SqZ is easily identifiable due to the microstructural difference. However, extent of 'splaying' in SqZ is far less than that of rhea eggshell and more similar to a 'cryptoprismatic' SqZ reported from non-avian maniraptoran eggshell *Macroelongatoolithus* (*Jin et al., 2007*). The grains in the EZ becomes weakly prismatic. Low-angle GB are mostly situated at SqZ as in rhea eggshell. In ML and EZ, GB are linear. In the SqZ, GB are rugged.

### Tinamous (*Eudromia elegans* and *Nothoprocta perdicaria*)

The eggshells of elegant-crested tinamou (*Eudromia elegans*) and Chilean tinamou (*Nothoprocta perdicaria*) are described together due to their similarity (*Figures 7 and 8*). The ML is characterized by clear needle-like calcite grains (*Zelenitsky and Modesto, 2003*; *Grellet-Tinner and Dyke, 2005*; *Grellet-Tinner, 2006*), which is reminiscent of that of non-avian maniraptoran eggshells (especially *Elongatoolithus* and *Reticuloolithus*) (*Figure 7—figure supplement 1*). The overall contour of ML is usually round (*Grellet-Tinner, 2006*). The boundary between the ML and SqZ is clear. In SqZ, the grain shape is highly irregular in elegant-crested tinamou eggshell, but Chilean tinamou eggshell has 'splaying' structure. In EZ, the grains are massive. Low-angle GB mostly exist in ML. In ML and EZ, GB are linear but in the SqZ, GB are highly rugged.

### Thin (possibly *Pachyornis geranoides*) and middle (possibly *Euryapteryx curtus*) thickness moa eggshells (Dinornithiformes)

These samples are most likely eggshells of *Pachyornis geranoides* and *Euryapteryx curtus*, respectively, although they may belong to a single species (see Appendix 1; *Gill, 2022*). The overall microstructure and crystallography of both eggshells are similar to those of rhea eggshell (*Figures 9 and 10*). The ML is composed of wedge-like calcite grains. The boundary between the ML and SqZ is clear. The SqZ is characterized by the 'splaying' of the grain shape. However, the EZ is not as prominent as that of rhea eggshell although GB is comparatively linear in the middle thickness moa eggshell (*Figure 10*). The grains in the EZ are irregular in shape, and this may be the reason why EZ of moa eggshell was not reported until the early 2000s notwithstanding the fact that moa eggshells had been described since the late 1800s (*Zelenitsky et al., 2002*). The GB features for thin and middle thickness moa eggshells are similar to that of rhea eggshell. The only difference is that the GB linearity at the EZ of thin and middle thickness moa eggshells is much weaker than that of rhea eggshell.

### Thick moa (*Dinornis novaezealandiae*) eggshells (Dinornithiformes)

This specimen is an unequivocal eggshell of *Dinornis novaezealandiae* (see Appendix 1; *Gill, 2022*). The microstructure is similar to that of ostrich eggshell in that long prismatic shell units occupy the

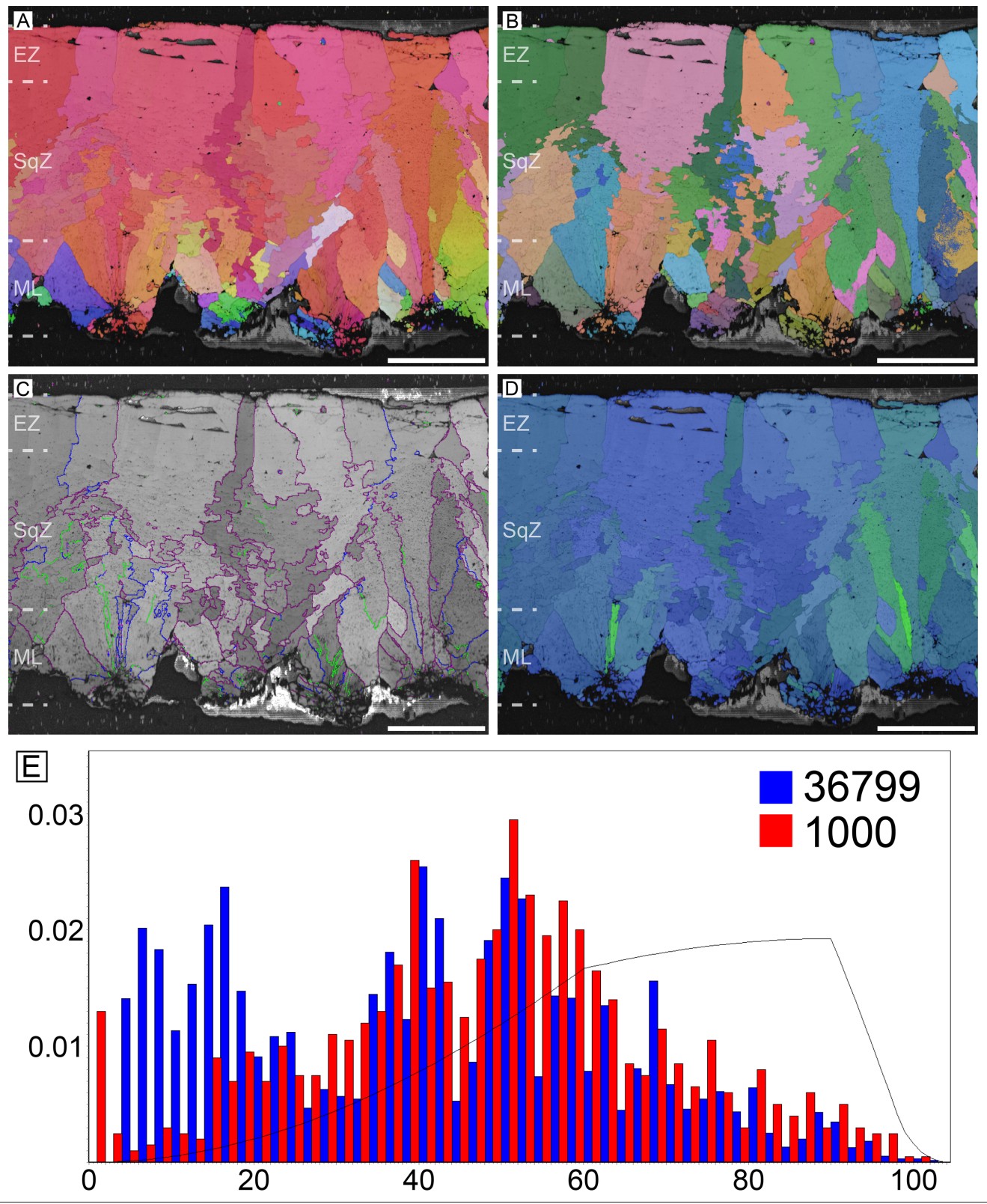

**Figure 5.** Kiwi eggshell. Scale bars equal 100 μm.

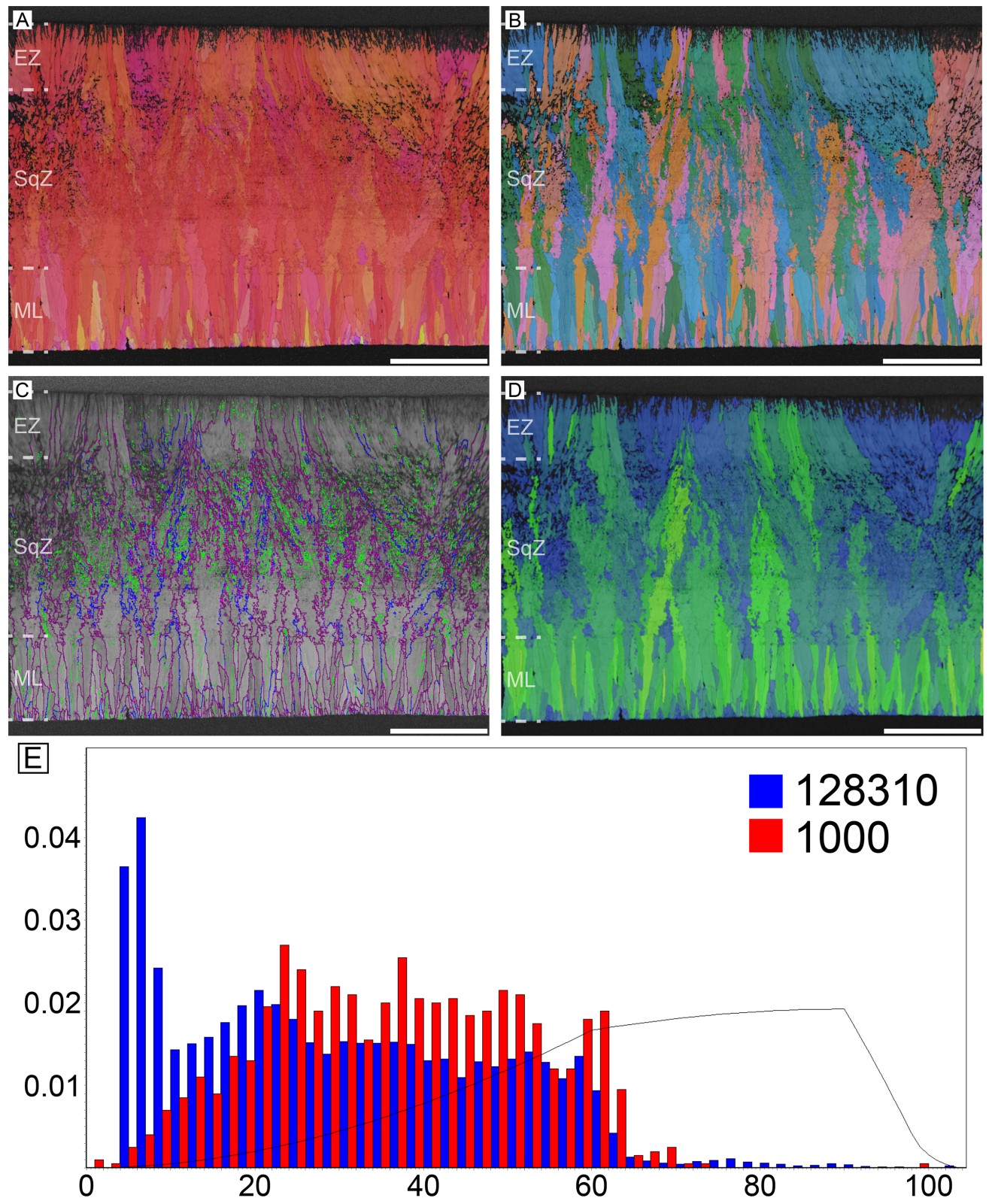

**Figure 6.** Elephant bird eggshell. Scale bars equal 1000 μm.

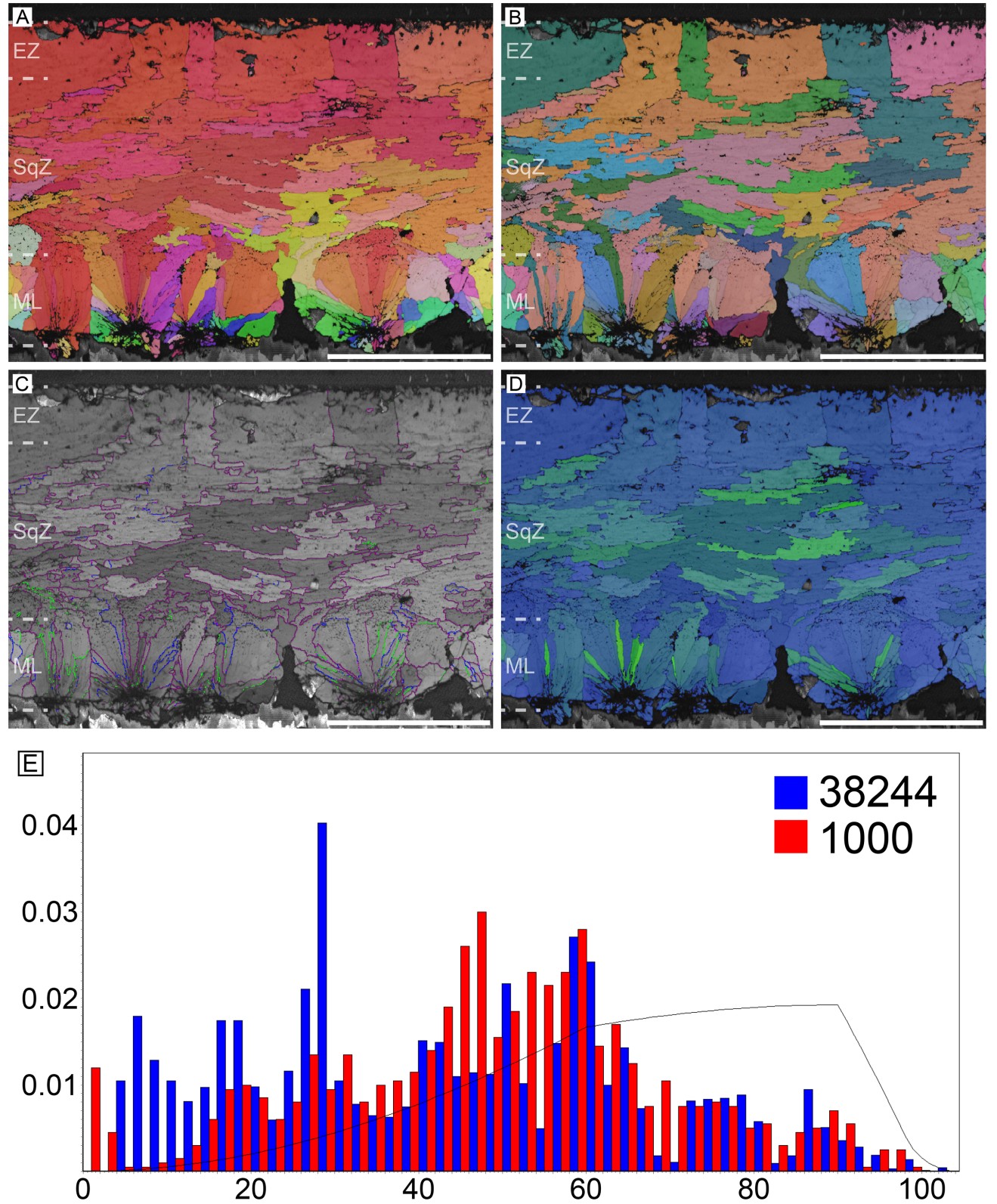

**Figure 7.** Elegant-crested tinamou eggshell. Scale bars equal 100 μm.

The online version of this article includes the following figure supplement(s) for figure 7:

**Figure supplement 1.** Mammillary layers of (**A**) *Reticuloolithus*, (**B**) *Elongatoolithus*, (**C**) tinamou, (**D**) kiwi, (**E**) thick moa, and (**F**) ostrich eggshells presented in AR, Euler, and GB mappings.

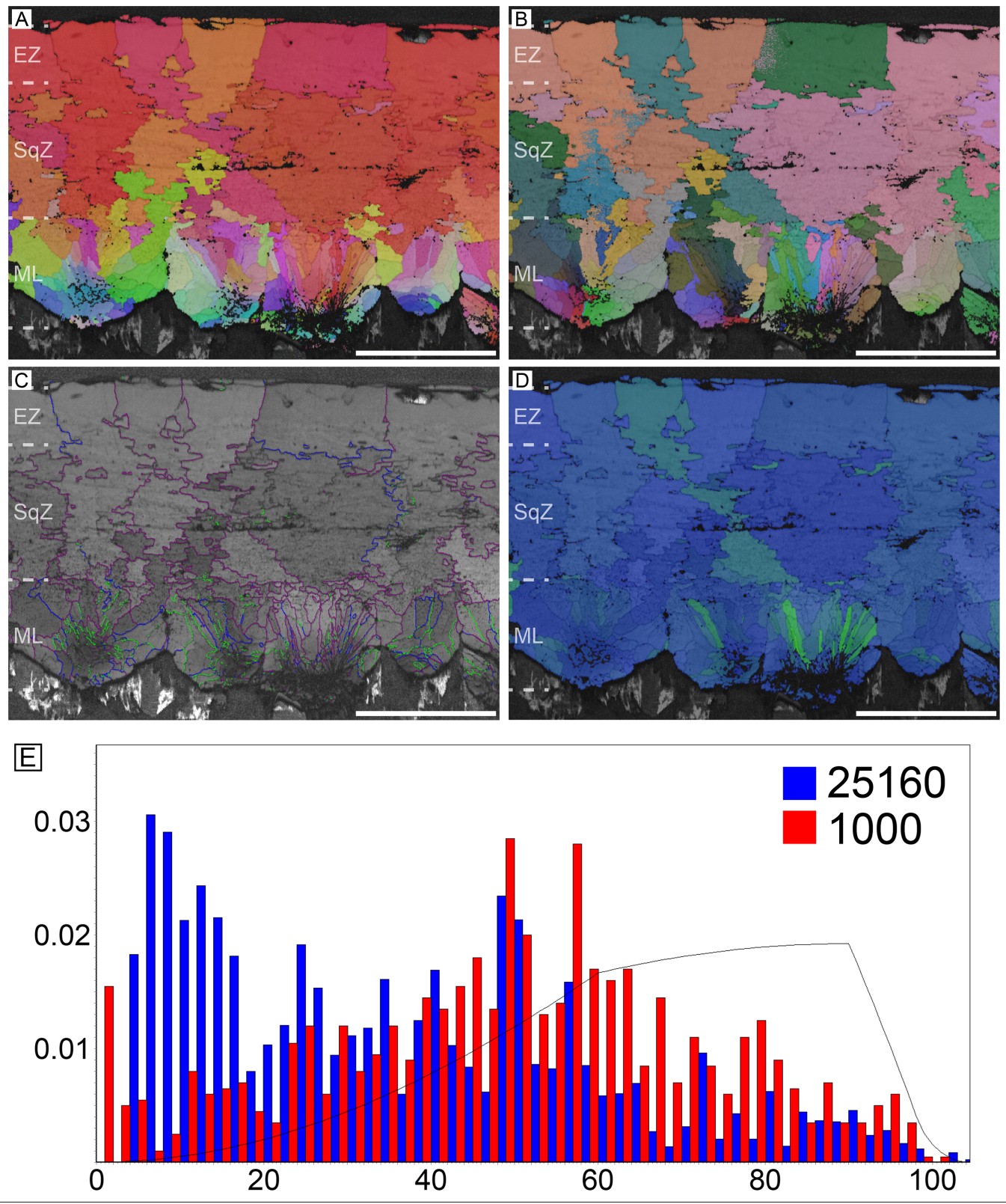

**Figure 8.** Chilean tinamou eggshell. Scale bars equal 100 μm.

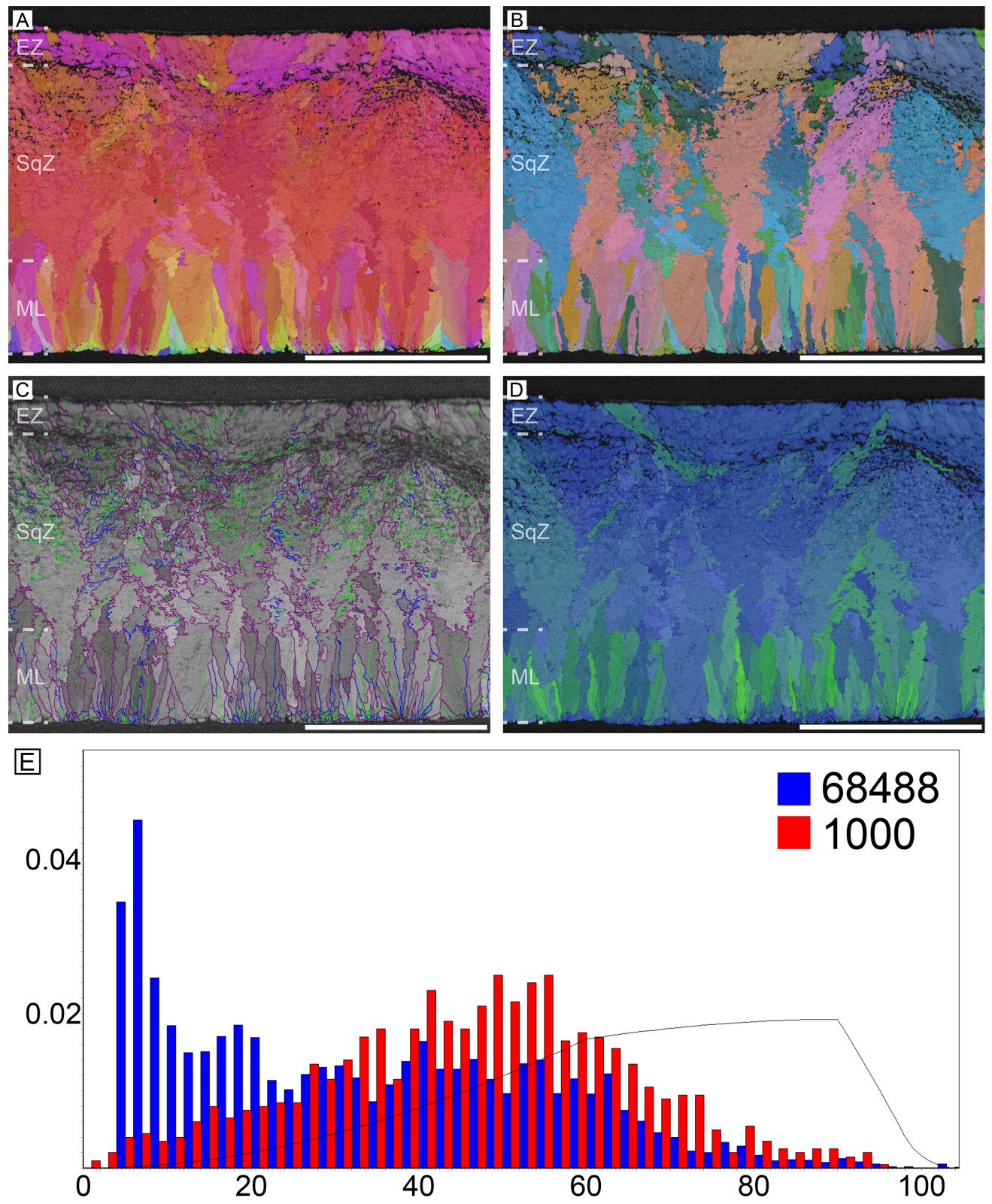

**Figure 9.** Thin moa eggshell (potential egg of *Pachyornis*). Scale bars equal 500 μm.

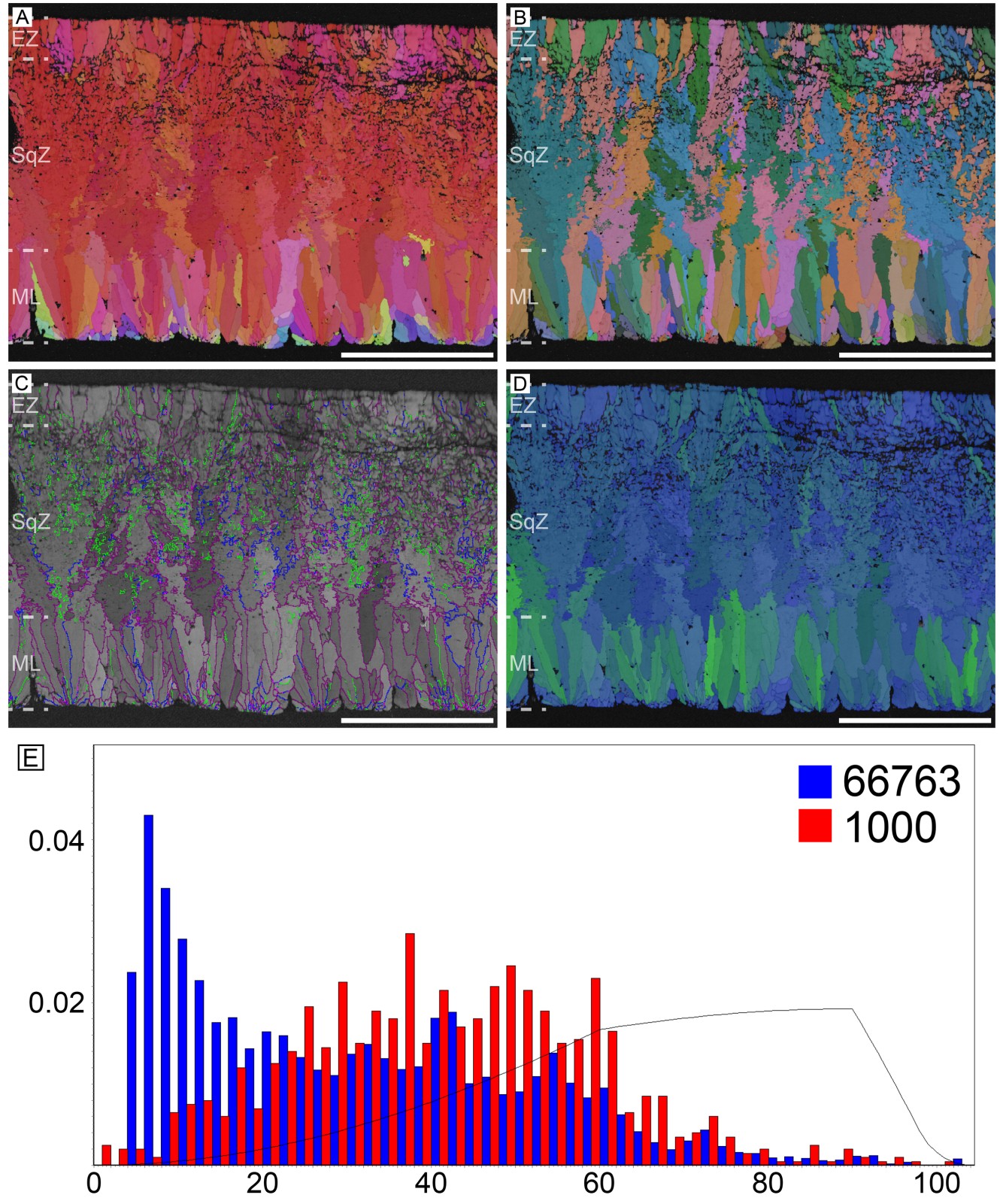

**Figure 10.** Middle thickness moa eggshell (potential egg of *Euryapteryx*). Scale bars equal 500 µm.

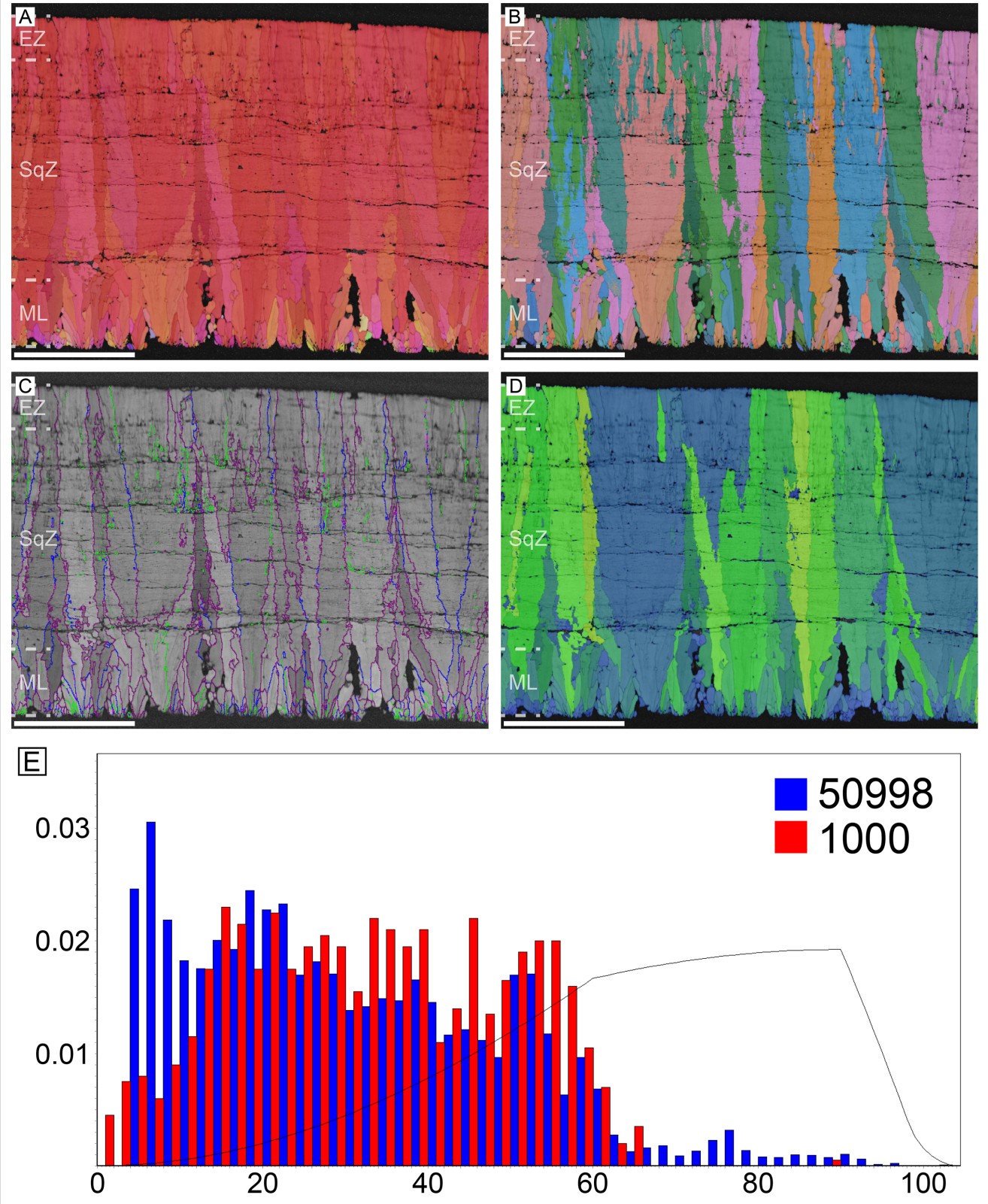

**Figure 11.** Thick moa eggshell (egg of *Dinornis*). Scale bars equal 500 μm.

whole thickness of the eggshell although the shell units are not as narrow as those of ostrich eggshell (*Figure 11*). The ML is wedge-like. It is worth mentioning that the outline of ML is round (*Figure 7—figure supplement 1*). The boundary between the ML and SqZ, SqZ and EZ are not clear because the 'splaying' of the SqZ is very weak as in ostrich eggshell. However, as in the case of ostrich eggshell, the GB condition provides an alternative way for identification of SqZ and EZ (*Figure 1—figure supplement 1*). Low-angle GB are not confined to a certain layer, but widespread in the eggshell. Similar to ostrich eggshell, GB are mostly linear and lack 'splaying' microstructure and highly rugged GB in SqZ. However, in magnified view, one can observe slight ruggedness in SqZ (*Figure 1—figure supplement 1*).

### Lithornis

Paleocene *Lithornis* eggshell has many features in common with tinamou eggshells (*Figure 12*; *Houde, 1988*; *Grellet-Tinner and Dyke, 2005*). The ML of *Lithornis celetius* eggshell is composed of needle-like calcite and overall shape of ML is weakly round in some parts of the eggshell. The SqZ shows clear 'splaying' structure and the crystals of EZ is massive. *Lithornis* eggshell is more similar to the Chilean tinamou eggshell (*Figure 8*) compared to the elegant-crested tinamou eggshell (*Figure 7*). Low-angle GB is mostly present in ML. ML and EZ are composed of linear GB, while SqZ is composed of rugged GB. The prominent slash patterns inside the SqZ are calcite twinning, which are diagenetically deformed calcite structure only found in fossil eggshells (i.e. abiogenic in origin; *Choi et al., 2021*). See also *Grellet-Tinner and Dyke, 2005* for SEM micrographs of *Lithornis vulturinus* eggshell that has wedge-like ML.

See also Appendix 3 and 4 for selected neognath and non-avian maniraptoran dinosaur eggshells for comparison.

## Overview for inverse pole figure mapping

Morphologically, palaeognath eggshells had been loosely categorized into three morphotypes (*Zelenitsky and Modesto, 2003*; *Grellet-Tinner and Dyke, 2005*; *Grellet-Tinner, 2006*). Ostrich-style (i.e. ostrich and thick moa eggshell) consists of wedge-like ML and prismatic shell units with near-absence of 'splaying' SqZ. Rhea-style (i.e. rhea, emu, cassowary, elephant bird, and thin & middle thickness moa eggshells) has been characterized by wedge-like ML and splaying SqZ. Finally, the tinamou-style (i.e. tinamou, kiwi, and *Lithornis* eggshells) is represented by needle-like ML, splaying SqZ, and massive EZ (but see below). Noticeable qualitative features (*Figure 13*) of palaeognath eggshells are: (i) calcite grains have strong vertical *c*-axis alignment (hence, mostly reddish in IPF Y mappings), which may be homologous to that of Mesozoic maniraptoran eggshells (*Moreno-Azanza et al., 2013*; *Choi et al., 2019*; *Choi et al., 2020*; *Choi et al., 2022*); (ii) ML is mostly composed of wedge-like calcite but needle-like calcite is present or dominant in tinamou-style eggshells (*Zelenitsky and Modesto, 2003*; *Grellet-Tinner and Dyke, 2005*; *Grellet-Tinner, 2006*; *Figure 7—figure supplement 1*); (iii) tinamou-style eggshells have round (or barrel-shaped) ML (*Grellet-Tinner, 2006*); thick moa eggshell appears to have round ML (*Figure 1—figure supplement 1*); (iv) ML and SqZ are easily differentiated due to grain shape differences except for in ostrich-style eggshells where the boundary between the two layers is unclear; (v) calcite in SqZ are mostly 'splaying' (sensu *Panhéleux et al., 1999*). However, calcite in SqZ of ostrich-style eggshells are nearly prismatic; (vi) EZ exists in all palaeognath eggshells; (vii) cassowary and emu eggshells have peculiar ornamentation on the outer surface ( = 'granular layer' sensu *Mikhailov, 1997a*) and very porous outer EZ (see *Figure 3—figure supplement 1*).

Compared to palaeognath eggshells, neognath eggshells are characterized by: (i) comparatively weak vertical *c*-axis alignment (see Appendix 3 for selected neognath eggshells; *Grellet-Tinner et al., 2017*; *Choi et al., 2019*; *López et al., 2021*; *Chiang et al., 2021*; see also *Oser et al., 2021* for a similar case of Mesozoic egg); (ii) boundary between ML and SqZ is not easily identified because of prismatic shell units (*Mikhailov, 1997a*). However, SqZ of common murre (*Uria aalge*) (Charadriiformes) eggshell is similar to that of rhea-style palaeognath eggshell in that 'splaying' is clear (*Appendix 3—figure 4*) and the overall structure of European green woodpecker (*Picus viridis*) eggshell (*Appendix 3—figure 5*) is remarkably similar to that of tinamou-style palaeognath eggshell. *Mikhailov, 2019* pointed out that some eggshells of four neognath clades (Galliformes, Anseriformes, Coraciiformes, and Piciformes) have palaeognath-eggshell-like microstructure and Charadriiformes might be added to this list (*Appendix 3—figure 4*).

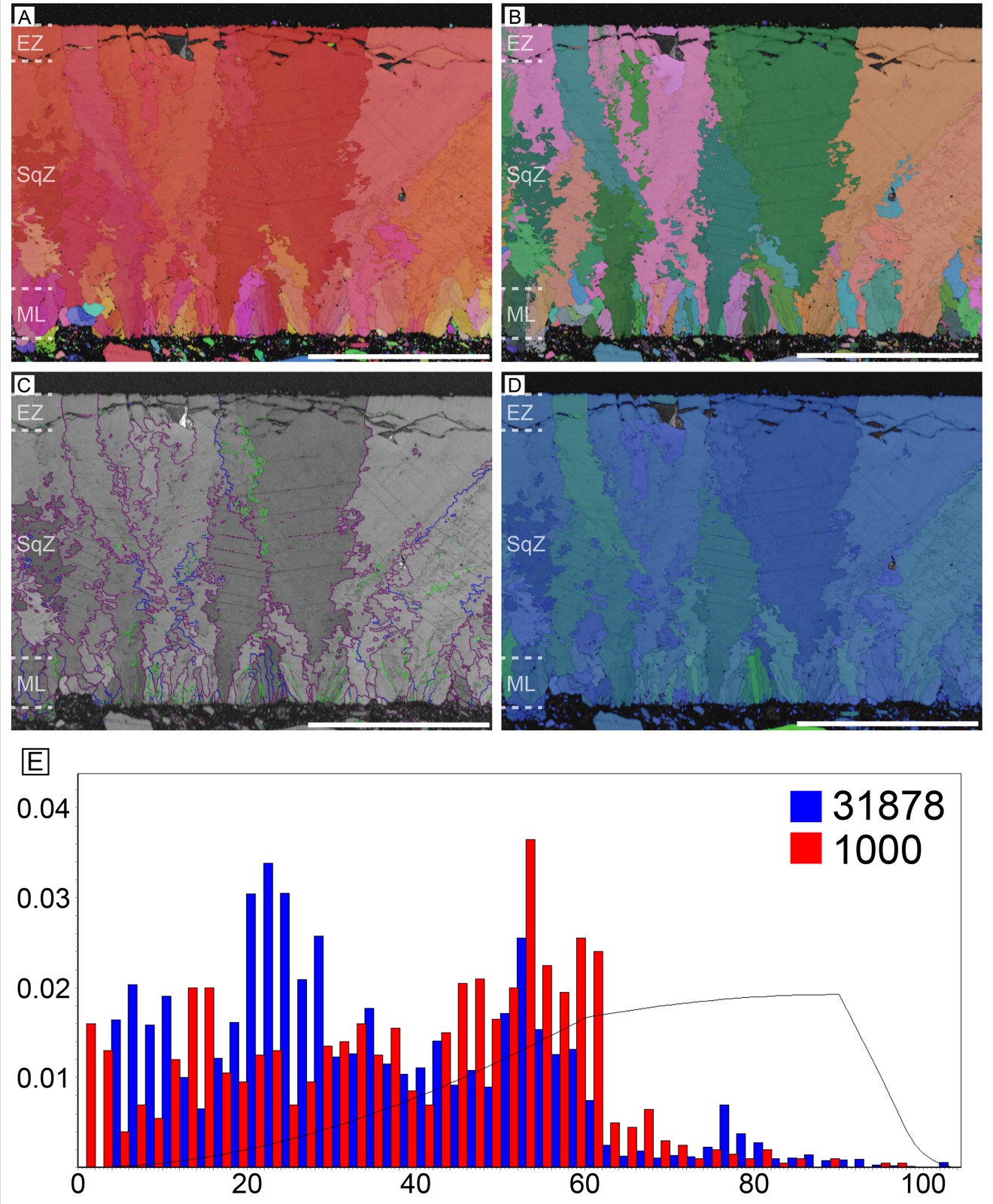

**Figure 12.** *Lithornis* eggshell (see also *Houde, 1988*; *Grellet-Tinner and Dyke, 2005*). Scale bars equal 250 µm.

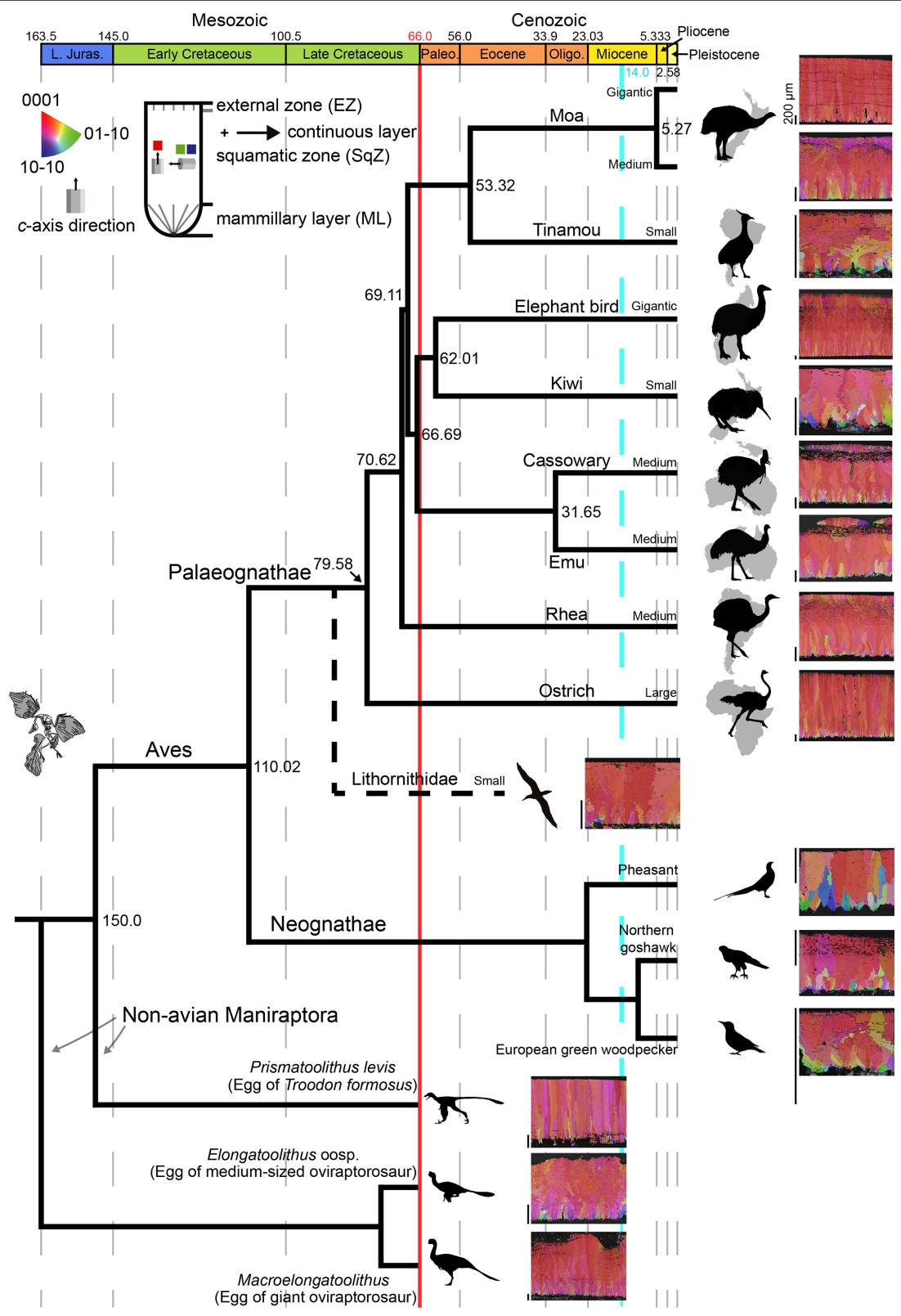

**Figure 13.** IPF mappings of palaeognath, neognath, and non-avian maniraptoran eggshells. The phylogeny and speciation timelines of palaeognath are based on *Yonezawa et al., 2017* and *Bunce et al., 2009*; branching in neognath and non-avian Maniraptora are arbitrary, not reflecting specific time. The speciation time of Lithornithidae (marked with a dashed line) is unknown. Silhouettes represent taxa and their habitats. All scale bars left to the IPF mappings is 200 µm. A red solid line marks K/Pg boundary; a dashed skyblue line denotes the initiation of cooling events in Miocene that might have

*Figure 13 continued on next page*

Figure 13 continued

caused gigantism of Palaeognathae (**Crouch and Clarke, 2019**). Silhouettes are attributable to (http://www.phylopic.org): Emily Willoughby (*Citipati*); Scott Hartman (*Paraves*), and Matt Martyniuk (*Gigantoraptor*). Other artworks are drawn by SC and NHK.

Non-avian maniraptoran eggshells (see Appendix 4 for selected examples) have strong vertical *c*-axis alignment as in palaeognath eggshells (**Moreno-Azanza et al., 2013**; **Choi and Lee, 2019**; **Choi et al., 2019**; **Choi et al., 2020**; **Choi et al., 2022**). Shell unit structure of oviraptorosaur eggshells is similar to that of rhea-style palaeognath eggshells except for needle-like ML and absence of EZ, whereas shell unit structure of troodontid eggshell is strikingly similar to ostrich-style palaeognath eggshell. See **Choi et al., 2019** for further information.

## Grain boundary mapping and actual size & thickness of egg

The main features of palaeognath eggshells are: (i) ostrich-style and rhea-style eggshells have extensive low-angle GB (lower than 20 degrees; green and blue lines in **Figure 14**) although the positions of high densities of low-angle GB vary in each clade. In ostrich eggshell, low-angle GB are concentrated at the outer part of ML. In rhea, elephant bird, and thin moa eggshells, low-angle GB are mostly concentrated at the SqZ. In emu and cassowary eggshells, low-angle GB are not widespread in SqZ, but abundant in the granular layer. In the thick moa eggshell, low-angle GB is not confined to certain positions; (ii) High-angle GB are dominant in tinamou-style eggshells and low-angle GB are mostly present in ML as in neognath eggshells; (iii) Ruggedness of GB changes abruptly at the boundary between SqZ and EZ in rhea, emu, kiwi, elephant bird, tinamou, and *Lithornis* eggshells although cassowary and thin moa eggshells show less prominent change. The ruggedness of GB is very slightly changed in ostrich-style eggshells, which have prismatic shell units.

In general, large eggs have thick eggshells and small eggs have thin eggshells (**Figure 14**; **Ar et al., 1979**; **Juang et al., 2017**). A notable outlier to this trend is kiwi eggs. Although kiwi eggs are large (especially compared to their body size; **Abourachid et al., 2019**), their eggshell is thin (**Vieco-Galvez et al., 2021**), comparable to that of smaller tinamou eggshell. Besides, the ellipticity and asymmetry of diverse avian eggs were investigated by **Stoddard et al., 2017**. We reused their data to present the egg shape indices of palaeognath eggs (**Figure 14—figure supplement 1**). The result shows that, compared to neognath eggs, palaeognath eggs are characterized by low asymmetry but ellipticity distribution is not very different from that of neognath eggs, consistent with the result of **Deeming, 2018**.

## Calcite grain aspect ratio

The main characteristics of palaeognath eggshells are (**Figure 15**): (i) rhea, emu, cassowary, kiwi, tinamou, thin moa, and *Lithornis* eggshells have relatively low aspect ratio (AR); (ii) ostrich, elephant bird, and thick moa eggshells show high AR. Compared to other palaeognath eggshells, these three eggshells have highly positively skewed AR distribution as well. Notably, these three eggshells are also the thickest among the palaeognath eggshells (**Figure 14**).

Neognath eggshells analysed in this study do not show high AR (Appendix 3). *Prismatoolithus levis* and *Triprismatoolithus* show high AR, while *Elongatoolithus* has low AR (Appendix 4). Intriguingly, *Macroelongatoolithus*, which was suggested to have 'cryptoprismatic' shell unit structure (**Jin et al., 2007**) shows intermediate AR between the two extremes (**Figure 15**).

Appendix 5 and **Figure 7—figure supplement 1** discuss how AR could be used to diagnose 'needle-like' calcite grains in ML.

## Misorientation distribution

**Choi et al., 2019** showed that low-angle (<20 degrees) are dominant in the misorientation distribution (MD) of ostrich and rhea eggshells whereas high-angle (>20 degrees) are dominant in MD of neognath eggshells analysed in that study. In this study, MD information of palaeognath eggshell is extended to all clades of Palaeognathae. Ostrich and rhea eggshells show low-angle dominant MD under neighbour-pair method (hereafter Type 1 distribution sensu **Choi et al., 2019**; **Figure 16**). This pattern is also present in elephant bird, thin, and middle thickness moa eggshell. Emu, cassowary, and thick moa eggshells show slightly different MD: low-angle is less well-dominant compared to the eggshells of the ostrich, rhea, elephant bird, and thin and middle thickness moa eggshells. In contrast,

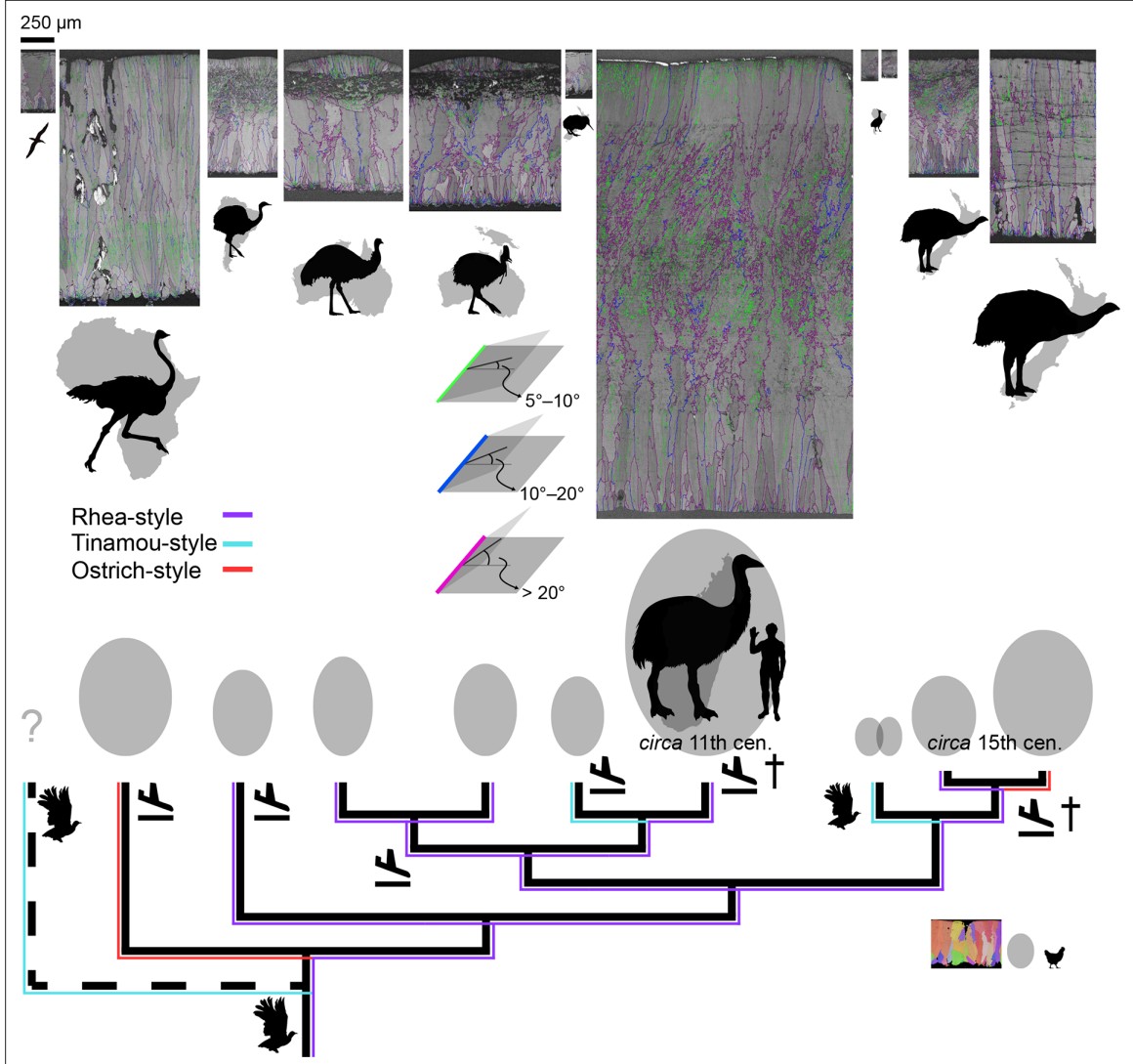

250 μm

Rhea-style
Tinamou-style
Ostrich-style

5°–10°

10°–20°

> 20°

*circa* 11th cen.

*circa* 15th cen.

**Figure 14.** GB mappings, eggshell thickness, and egg size of Palaeognathae. The green, blue, and purple lines in GB mapping denote the angle range between the calcite grains. All eggshell maps (including IPF mapping of chicken eggshell for comparison) are drawn to scale; note a scale bar at the upper left corner. The silhouettes of palaeognath are drawn to scale (note a human next to elephant bird and a chicken at the lower right corner). Egg shape and size are drawn to scale (*Hauber et al., 2014*; *Stoddard et al., 2017*). Two recently extinct lineages are marked by daggers and the extinct Lithorinithidae by a dashed branch. Landing symbols denote potential independent losses of flight (*Mitchell et al., 2014*; see also *Sackton et al., 2019*) and flying bird silhouettes denote volant taxa. Sky blue lines show microstructural and crystallographic similarities among tinamou-style eggshells that is attributable to homoplasy. Red lines mean the homoplastic similarities between ostrich-style eggshells. Purple lines represent potential homologies of rhea-style eggshells.

The online version of this article includes the following figure supplement(s) for figure 14:

**Figure supplement 1.** Ellipticity and asymmetry of palaeognath eggs.

**Figure supplement 2.** An alternative interpretation of evolution of palaeognath eggshells assuming that all tinamou-style eggshells are homologous to one another.

eggshells of kiwi, tinamou, and *Lithornis* have more high-angle dominant MD. The MD patterns of palaeognath eggshells are more diverse than previously postulated by *Choi et al., 2019*.

Neognath eggshells used in this study showed high-angle (>20 degrees) dominant MD, consistent with the result of *Choi et al., 2019* (hereafter, Type 2 distribution sensu *Choi et al., 2019*; *Figure 16*). As far as we know, there is no neognath eggshell that has Type 1 distribution. Even though microstructure of common murre eggshell is similar to that of rhea-style palaeognath eggshell, it does not have Type 1 distribution (*Appendix 3—figure 4*).

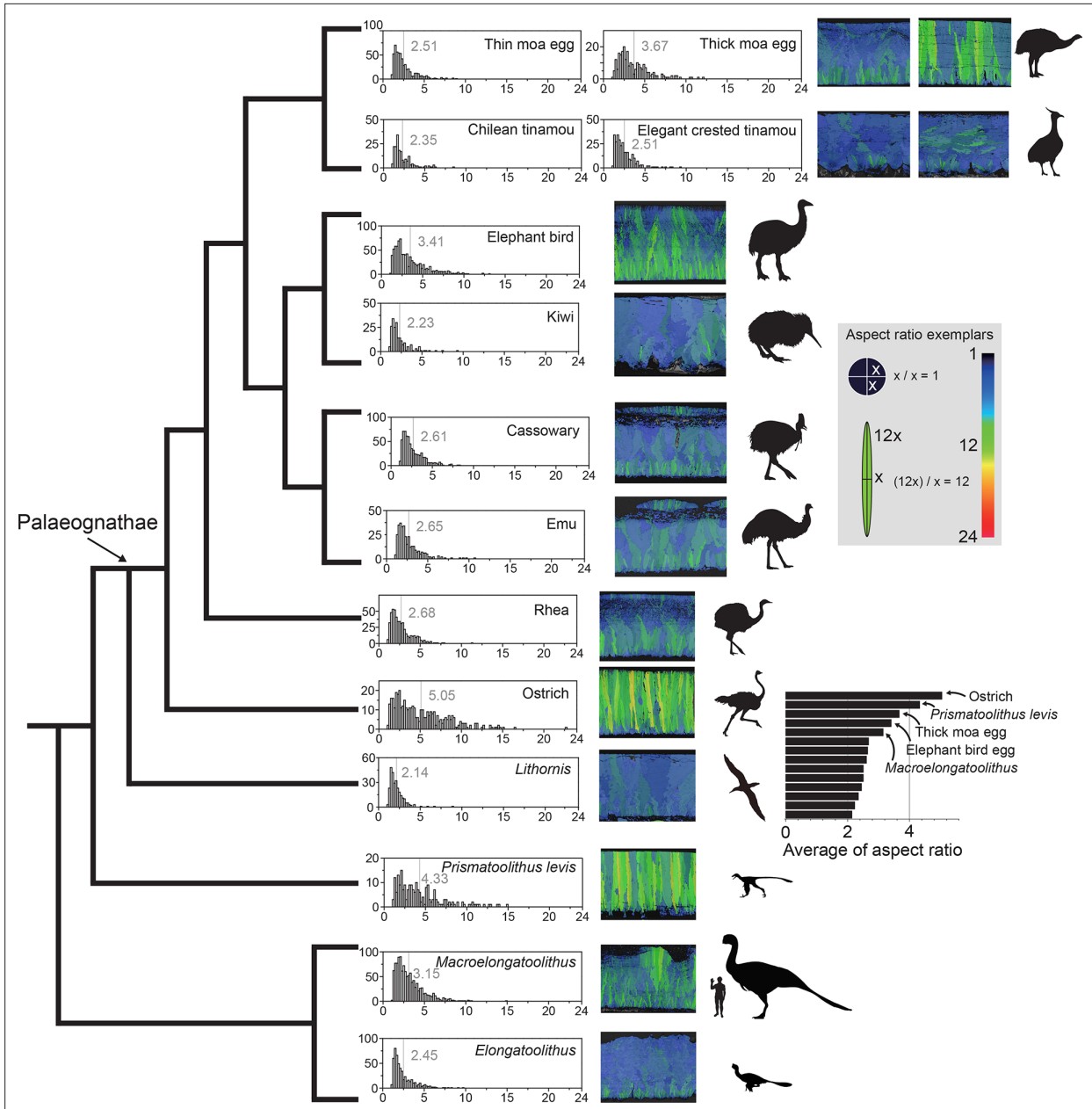

**Figure 15.** AR mappings and histograms of palaeognath and non-avian maniraptoran eggshells. Note that ostrich, (Late Cretaceous) *Prismatoolithus levis*, giant moa (*Dinornis*), elephant bird, and (Late Cretaceous) *Macroelongatoolithus* eggshells are characterized by higher AR. The vertical bars and numbers in the histograms mean the average point of AR distribution and its value, respectively. Silhouettes of non-avian dinosaur are drawn to scale.

As discussed in *Choi et al., 2019*, Type 1 and 2 distributions already existed in Cretaceous non-avian maniraptoran eggshells (*Figure 16*; see also *Moreno-Azanza et al., 2013*; *Choi and Lee, 2019*; *Choi et al., 2020*; *Choi et al., 2022*).

## Ancestral state reconstructions

Ancestral states for mean of neighbour-pair MD are very similar for both phylogenetic trees (*Yonezawa et al., 2017*; *Kimball et al., 2019*), with a relatively constant ancestral value (~32°) for several major palaeognath clades (Palaeognathae, Notopalaeognathae, Novaeratitae – clade names sensu *Sangster et al., 2022*; *Figure 16*). A conspicuous increase is observed for both Casuariiformes (34.7° [Kimball] or 35.3° [Yonezawa]) and Tinamiformes (35.5° [Kimball] or 37.2° [Yonezawa]), while Dinornithiformes show a slight decrease (31.2° [Kimball] or 31.4° [Yonezawa]). The Apterygiformes-Aepyornithiformes

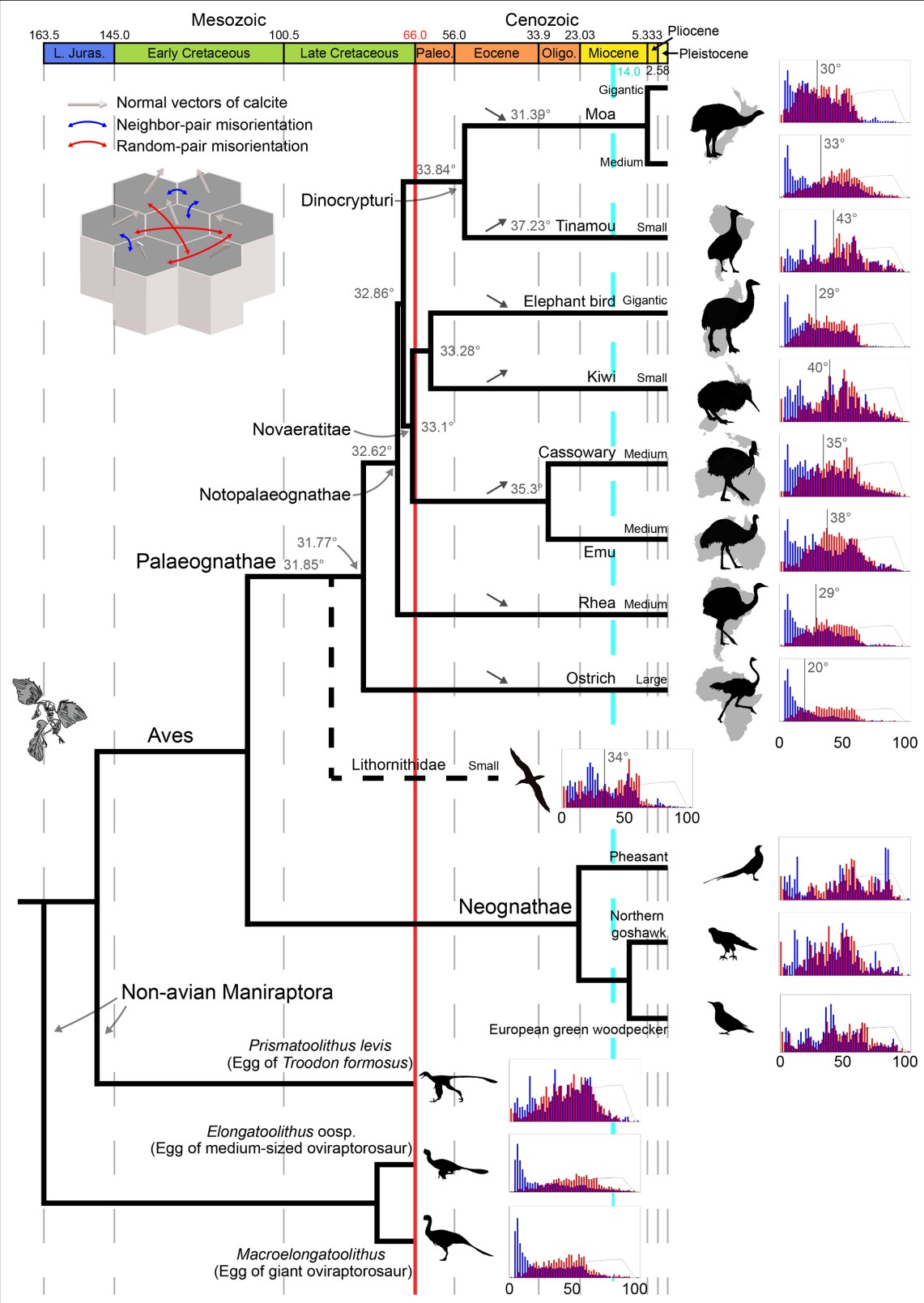

**Figure 16.** Misorientation distributions of palaeognath, neognath, and non-avian maniraptoran eggshells (adapted from *Figure 13*). The vertical bars and numbers in the histograms mean the average point of neighbour-pair misorientation (blue) and its value, respectively. The numbers at the nodes represent the ancestral states for mean of neighbour-pair MD (note that only one MD of tinamou eggshell was shown for brevity). Up and down arrows mark the changing ancestral state trends of each node compared to the nearest ancestral states.

clade shows only a minor increase (33.3° for both trees), reflecting the divergence between its two sampled members (high value [40.4°] for *Apteryx*, low value [29.5°] for *Aepyornis*). In Novaeratitae and Dinocrypturi (i.e. moa +tinamou), the lowest values are observed in elephant bird and moa, potentially reflecting a low-angle trend associated with gigantism within that clade. The fact that high values are found for both small (tinamous, kiwi) and large (emu, cassowary) taxa suggests that the reverse is not true, although a larger sample size would be necessary to test that hypothesis. The ostrich shows a very low value (20.1°) compared to other Palaeognathae, suggesting a distinctive crystalline structure within its eggshell. This low value in the ostrich, however, does not affect the ancestral state at the Palaeognathae node in either tree – likely due to the inclusion of *Lithornis*, the earliest-diverging taxon in our sample, which shows a misorientation value of 34.0° closer to that of the recovered ancestral state for Palaeognathae (~32°).

Ancestral states for AR on the tree from *Yonezawa et al., 2017* do not exhibit any conspicuous pattern for major clades, with all internal nodes showing very similar values (~2.6). This is likely due to the small range in values for terminal taxa (2.14–5.05, with all but three species within a 2.14–3 range; *Figure 15*), which might also explain the very low phylogenetic signal for that trait. The distribution of AR among Palaeognathae seems to be correlated with that of body mass: the highest values (>3.4) are observed in large species (>100 kg – *Struthio*, Aepyornithidae, and *Dinornis*); small species (<10 kg – *Apteryx*, *Eudromia*, *Nothoprocta* and *Lithornis*) show lower values (<2.5); and medium-sized species (*Rhea*, *Dromaius,* and *Casuarius*) present an intermediate AR. The only exceptions are the two smaller moa species, which show a much smaller AR (*Pachyornis*: 2.51; *Euryapteryx*: 2.34) than expected for their body mass, comparable to that of tinamous. The two clades (*Eudromia*, *Nothoprocta*) and (*Pachyornis*, *Euryapteryx*) are the only subclades recovered with an ancestral AR under 2.6. This could potentially reflect a synapomorphy of calcite grain structure in Dinocrypturi, albeit not recovered for this small sample due to the high value in *Dinornis*. The ostrich is recovered as a clear outlier with the highest value in the sample (5.05), again supporting a highly autapomorphic crystalline structure in its eggshell.

## Discussion
### Evolution of the palaeognath eggshells through time

The phylogeny of Palaeognathae has experienced a set of revolutionary changes since 2008 (compare *Livezey and Zusi, 2007* and *Yonezawa et al., 2017*) and it provides an unexplored chance to trace the evolution of microstructure and crystallography of modern dinosaur eggshells. We interpret our results following the phylogeny of *Yonezawa et al., 2017*, which provides estimated speciation time-lines. But it should be noted that *Cloutier et al., 2019* and *Sackton et al., 2019* report an alternate phylogeny, which are characterized by a switching of the positions of rhea and (tinamou +moa). See Appendix 6 for interpretation based on the phylogeny of *Cloutier et al., 2019* and *Sackton et al., 2019*.

i.   Rhea-style microstructure would be synapomorphic to all Palaeognathae or more inclusive monophyletic group of birds (*Figure 14*) considering the presence of rhea-style eggshells in the Neognathae (see above; see also *Mayr and Zelenkov, 2021* for skeletal similarities between the Neognathae and a potential stem group Struthioniformes) and the presence of supposedly rhea-style eggshells in the Upper Cretaceous deposits (see below). Alternatively, Palaeogene *Lithornis* eggshell may represent the synapomorphic microstructure of Palaeognathae. Volant tinamou and *Lithornis* (*Altimiras et al., 2017*; *Torres et al., 2020*) share considerable similarity not only in their skeletal characters (*Houde, 1988*; *Nesbitt and Clarke, 2016*), but also in their eggshell microstructure (*Houde, 1988*; *Grellet-Tinner and Dyke, 2005*) and crystallography (this study). However, if we assume that similar microstructure of tinamou and *Lithornis* eggshells are indeed homology, then its corollary is that very similar rhea-style microstructure evolved from tinamou-style microstructure independently at least four times (*Figure 14—figure supplement 2*). Considering the remarkable microstructural similarities among rhea, casuariid, elephant bird, and moa eggshells, a more reasonable interpretation is that rhea-style eggshells are homologous and the similarity between tinamou and *Lithornis* eggshells is homoplastic (*Figure 14*). Indeed, the similarity between *Lithornis* and tinamou eggshells is not as complete as that of rhea-style eggshells, and the former similarity may be better described as 'incomplete convergence' (sensu *Herrel et al., 2004*; see also *Benito*

*et al., 2022* for an example of potential 'incomplete convergence' of palatal morphologies of Palaeognathae and non-avian theropod dinosaurs).

ii. The three tinamou-style eggshells (tinamous, kiwi, and *Lithornis*) would be homoplastic because the tinamou-style of kiwi would be autapomorphic as well. In fact, although *Zelenitsky and Modesto, 2003* observed acicular ML in kiwi and tinamou eggshells, the EBSD image of kiwi eggshell shows that wedge-like calcite is more dominant in ML (*Figure 5*) and this view is in agreement with the view of *Grellet-Tinner, 2006*. Thus, the morphological cohesiveness of 'tinamou-style' is not as solid as that of rhea-style eggshell, so we would suggest that the loosely categorized 'tinamou-style' (e.g. *Grellet-Tinner and Dyke, 2005*) should not be understood as homologous entity because it is an oversimplification of morphological variability in tinamou, kiwi, and *Lithornis* eggshells (e.g. *Benito et al., 2022*).

The homoplasy interpretation of tinamou and *Lithornis* eggshells brings about an unresolved question. Among the mostly flightless Palaeognathae, tinamou and *Lithornis* are capable of flight (*Figure 14*). *Legendre and Clarke, 2021* showed that flight affects eggshell thickness, but the authors suggested that whether flight affects microstructures of avian eggshells should be further investigated. *Mitchell et al., 2014* proposed that there were at least six loss of flight events (*Figure 14*) in palaeognath lineages (see also *Sackton et al., 2019*). If so, most of the loss of flight events did not cause a transition in eggshell microstructure (i.e. rhea, Casuariidae, elephant bird and small-to-medium sized moa). Ostrich, kiwi, and thick moa eggshells are exceptions to this trend although there is possibility that ancestral flightless ostrich, kiwi, and large moa eggshells might have rhea-style microstructure in the first place. This hypothesis can be only testable through (unexplored) fossil record.

In this scenario, volant tinamou and *Lithornis* acquired roughly similar microstructure, although they maintained flight (*Figure 14*), meaning that maintaining or abandoning of flight had little influences on microstructures. Instead, the exotic microstructure of tinamou might be related to their cladogenesis (*Almeida et al., 2022*) because thin and middle thickness moa (a sister clade of tinamou; *Phillips et al., 2010*) eggshells maintain rhea-style microstructure. Similarly, the microstructure of kiwi eggshell (see also *Vieco-Galvez et al., 2021*) would be autapomorphic considering the rhea-style microstructure of its sister clade elephant bird (*Mitchell et al., 2014*). Considering that ancestral kiwi might have been a volant clade (*Worthy et al., 2013*), the disproportionally large size of egg and peculiar microstructure of kiwi might have appeared when the cladogenesis of (flightless and extremely precocial) Apterygidae took place (*Worthy et al., 2013*).

If cladogenesis is indeed related to the evolution of autapomorphic microstructures (see also ostrich-style and casuariid eggshells below), it may have implication for Lithornithidae monophyly (*Widrig and Field, 2022*). The monophyly of Lithornithidae is supported by recent views (*Nesbitt and Clarke, 2016*; *Yonezawa et al., 2017*) but there are also different results that support lithornithid paraphyly (*Houde, 1988*; *Worthy et al., 2017*). Compared to other clades of Palaeognathae, tinamou and kiwi are speciose (*Weir et al., 2016*; *Almeida et al., 2022*). If all members of respective group share similar microstructure, it may mean that autapomorphic microstructure may be a feature of monophyletic group. Then, the autapomorphic microstructure of *Lithornis* eggshell may indirectly support lithornithid monophyly rather than paraphyly.

iii. Homoplasy-based explanation can be also applicable to the similarity between the two ostrich-style eggshells. Ostrich have evolved their peculiar eggshell microstructure after the split from all other Palaeognathae in the Late Cretaceous (*Figure 13*). In case of moa, the cladogenesis of *Dinornis* (ostrich-style eggshell) and other moa (rhea-style eggshells) might have happened in 5.27 Ma (*Bunce et al., 2009*). Thus, considering the phylogenetic topology of Palaeognathae, the similarity between ostrich and thick moa eggshells was not derived from the common ancestry, therefore, it is homoplasy (*Figure 14*).

iv. Casuariid (emu and cassowary) eggshells are nearly identical. We interpret that their MRCA that lived in Palaeogene Australia (a flightless bird most likely adapted to vegetated and very humid habitat as modern cassowary does; *Moore, 2007*; *Worthy et al., 2014*; *Mitchell et al., 2014*) had already acquired this unique microstructure. Cassowary still lives in humid environments of New Guinea and northeastern Australia, but emu is widely distributed in Australia, including arid environments (*Langley, 2018*). The role of peculiar microstructure in casuariid eggshells in drastically different environments should be further investigated unless emu have maintained their microstructure simply due to 'phylogenetic inertia' effect (*Edwards and Naeem, 1993*; but see also *Shanahan, 2011*). The similarity between emu and cassowary eggshells would be homology.

v. The rate of evolutionary change in microstructures of eggshell varies among clades. Casuariid eggshells show long stasis; eggshells of both emu and cassowary changed little since

their speciation (31.65 Ma; *Figure 13*). In contrast, moa eggshells imply a very different story. *Bunce et al., 2009* stated that large and medium sized moa diverged in the Pliocene (5.27 Ma; *Figure 13*). This is not necessarily a long time-interval in evolutionary biology and palaeontology, but moa eggshells show very different microstructures and crystallography (*Figure 13*). The contrasting stubborn conservatism (Casuariidae) and swift change (moa) in eggshell microstructure may mean that phenotypic evolution of microstructure of eggshell might not be gradual (e.g. *Gould and Eldredge, 1993*; see also *Pennell et al., 2014b*).

vi. Rhea- and ostrich-style eggshells have Type 1 distribution of MD (or weakened Type 1) while tinamou-style eggshells have Type 2 distributions of MD, therefore, MD pattern of palaeognath eggshells are more complicated than the one postulated by *Choi et al., 2019*. *Choi et al., 2019* posed two scenarios to infer the ancestral MD of Neornithes: the first hypothesis assumed that Type 1 distribution of Palaeognathae is directly inherited from non-paravian dinosaurs; the second hypothesis assumed that Paraves acquired Type 2 distribution while Palaeognathae re-evolved Type 1 distribution. However, both hypotheses assumed that the ancestral state of MD of Palaeognathae is Type 1 although *Choi et al., 2019* analysed just ostrich and rhea eggshells. According to the ancestral state reconstruction presented in this study, it is highly likely that early-diverging Palaeognathae had the weak Type 1 MD notwithstanding the fact that volant *Lithornis* eggshell is characterized by Type 2 MD (*Figure 16*). Unless this view is negated by future findings, currently, our results show that the two hypotheses posed by *Choi et al., 2019* are based on valid postulation (Type 1 MD for ancestral state), which raises additional scientific questions (see below).

We note that our interpretation is mainly based on the phylogeny of *Yonezawa et al., 2017*, but that might not be the final consensus on this issue (e.g. *Sackton et al., 2019*). Hence, the interpretation of palaeognath eggshell evolution should depend on the ongoing advancements of palaeognath phylogeny and should be updated accordingly (e.g. agreement on the topology of tree, revised timelines of evolution, inclusion of new fossil taxa data).

## Implications to palaeontology

Palaeognath eggshells provide useful insights into palaeontology (*Figures 13, 15 and 16*) as a modern analogue.

i. Similar-looking eggshells can be laid by closely related taxa (e.g. Casuariidae). However, closely-related taxa can lay very different eggshells (e.g. differing eggshell among moa taxa). In palaeontology, an embryo of non-avian dinosaur *Troodon formosus* was found in an egg named *Prismatoolithus levis* (*Varricchio et al., 2002*). This taxon-ootaxon relationship has been widely (or over-widely) accepted that many prismatoolithid eggs were recognized as troodontid eggshell but it should be used with caution (see *Mikhailov, 2019*). There is possibility that at least some troodontid dinosaurs might have laid eggshells dissimilar to *P. levis* as in the case among moa species. On the other hand, distantly related non-avian dinosaur taxa might have laid similar-looking eggshells independently as in the case of ostrich and thick moa eggshells. These palaeognath eggshells are also morphologically very similar to eggshell of *Troodon formosus* (*Zelenitsky and Hills, 1996*; *Zelenitsky et al., 2002*; *Varricchio and Jackson, 2004*), another clear case of homoplasy (*Figure 15*).

ii. Differentiating homology from homoplasy in similar-looking phenotypes should have paramount importance in morphology-based fossil egg palaeontology (e.g. *Choi et al., 2020*). It is highly likely that prismatic microstructures of *P. levis*, ostrich, and thick moa eggshells are the outcome of homoplastic evolution. In palaeontology, many different types of eggshells are assigned to the oofamily Prismatoolithidae because they have prismatic shell unit structure, but it might be composed of eggshells from polyphyletic egg-layers. For example, if modern ostrich eggshells and thick moa eggshells are parataxonomically classified solely based on morphological criteria, they may be classified as 'Prismatoolithidae' although (moa +ostrich) is not a monophyletic group. Unless homoplastic characters are appropriately separated, the endeavors of parataxonomic systematics would have little evolutionary biological values but merely limited to morphological classification (*Mikhailov, 2014*; *Mikhailov, 2019*; see also *Varricchio and Jackson, 2004*), which is sometimes vulnerable to homoplasy (*Livezey and Zusi, 2007*; *Yonezawa et al., 2017*). In addition, for a better understanding of taxon-ootaxon relationship, homoplastic and homologous similarities should be clearly separated (e.g. see case of the Tuştea Puzzle; *Grigorescu, 2017*; *Botfalvai et al., 2017*). *McInerney et al., 2019* showed that the syrinx, hyoid, and larynx of Palaeognathae are less prone to homoplastic evolution, thus, they might be more valuable for morphology-based classification of

Palaeognathae. Similarly, future eggshell studies may concentrate on finding less-homoplasy-prone morphological entities of eggshells.

iii. Prismatic microstructure might be derived from rhea-style microstructure. Among non-avian maniraptoran eggshells, *Elongatoolithus* exhibits a rhea-style microstructure but *Prismatoolithus* presents an ostrich-style one (*Figure 13*). Intriguingly, the eggshell of gigantic oviraptorosaur (*Pu et al., 2017*) *Macroelongatoolithus* has an intermediate AR between *Elongatoolithus* and *Prismatoolithus* (*Figure 15*). It may represent the intermediate stage between the two morphotypes that might be related with the gigantism of oviraptorosaur. Although correlation is not very clear, we would like to emphasize that thick eggshell of ostrich and large moa are characterized by ostrich-style microstructure (hence, high AR) and the thickest elephant bird eggshell also has high AR (*Figure 15*). Investigating the relationship between the egg size, eggshell thickness, and AR of extinct maniraptoran eggshell from more future findings may provide further insight into the evolution and function of eggshell microstructure.

iv. There are thin 'ratite-morphotype' fossil eggshells from the Upper Cretaceous deposits (discussed in *Choi and Lee, 2019*). Considering the estimation that Palaeognathae and Neognathae diverged in the Early Cretaceous (*Lee et al., 2014*; *Yonezawa et al., 2017*; *Figure 13*), at least some of the Late Cretaceous 'ratite-morphotype' eggshells might belong to early-diverging (and volant) Palaeognathae. For example, the European ootaxa *Sankofa pyrenaica* (*López-Martínez and Vicens, 2012*), *Pseudogeckoolithus* cf. *nodosus*, and *P.* aff. *tirboulensis* (*Choi et al., 2020*) have remarkable rhea-style microstructure. Although, here again, the possibility of homoplasy should not be overlooked, further studies on Cretaceous materials may provide new indirect evidence on the presence of Palaeognathae in the Cretaceous. In fact, the presence of Neognathae in the Late Cretaceous was confirmed by body fossils from Maastrichtian (Late Cretaceous) deposits in Antarctica (*Clarke et al., 2005*) and Europe (*Field et al., 2020*), indirectly supporting the presence of Palaeognathae in the Late Cretaceous. If some Late Cretaceous rhea-style 'ratite-morphotype' eggshells turn out to be true palaeognath eggshells, our interpretation (*Figure 14*) will be further supported with evidence.

v. The ancestral state reconstruction of MD exemplifies the importance of fossils in ancestral reconstructions, especially when focusing on early nodes with a high discrepancy between extant and extinct species (e.g. *Finarelli and Flynn, 2006*; *Li et al., 2008*; *Cascini et al., 2019*; *Soul and Wright, 2021*). *Maddison et al., 1984* pointed out that, ideally, at least two outgroups are necessary to unambiguously polarize characters of ingroup taxa. In our study, the only outgroup for extant palaeognath eggshells is *Lithornis* eggshell. However, at least some rhea-style fossil eggshells from the Upper Cretaceous deposits (e.g. *Pseudogeckoolithus*) are characterized by Type 2 MD, which is also observed in *Troodon* and enantiornithine eggshells (*Figure 16*; *Choi et al., 2019*; *Choi et al., 2020*). If these rhea-style eggshells are confirmed as true palaeognath eggshells and can be included in the future ancestral reconstruction analysis, the ancestral state interpretation might be affected. With an additional outgroup down the phylogenetic tree of Palaeognathae, a better interpretation would be possible.

vi. The current parataxonomic classification usually used by palaeontologists is a compromise between the Linnean rank system (e.g. oofamily. oogenus, and oospecies; *Mikhailov et al., 1996*) and Hennigian cladistic approach (e.g. *Varricchio and Jackson, 2004*; *Grellet-Tinner et al., 2006*; *Zelenitsky and Therrien, 2008*). It is similar to a philosophy of evolutionary taxonomists who asserted that classification should find a balance between the overall similarity and genealogical history (*Wiley and Lieberman, 2011*, p. 3). They defined groups based on criteria (e.g. diagnosis) rather than common ancestry. We agree that naming a fossil egg with binomial nomenclature and diagnosis has clear merits for stratigraphic purposes and communications among researchers (*Mikhailov, 2014*). However, to guarantee the objectivity and reproducibility of the classification, we also agree that cladistic approach should be preferred over somewhat arbitrary similarity-based classification.

Nevertheless, the current cladistic approach for palaeoology is not without weaknesses. When a character of two or more different egg types is similar, they are coded into a same state (e.g. '0' or '1'). The presumption of this step is that the shared character state is a shared homolog (*Wiley and Lieberman, 2011*). Again, without the assurance that the same character state is not a homoplasy, the presumption can collapse, and the resultant cladogram can be a 'contaminated' result. That being said, homoplasies can be still useful for ootaxonomy because homoplasies may separately contribute to defining two or more monophyletic ootaxa. *Wiley and Lieberman, 2011* (p. 119) stated "… some homoplasies, taken together, are homoplastic; but taken separately, each may be independent taxic homologies of the monophyletic groups with which they are associated as a diagnostic property". In palaeognath eggshells, for example, roughly defined 'prismatic shell units' of ostrich and thick moa eggshells are homoplastic. However, if the similar features of 'prismatic shell units' of both eggshells are used as

their respective synapomorphy, the homoplasies will become new respective synapomorphies of the members of a formerly 'polyphyletic group' (i.e. ostrich +moa).

## Future research suggestions

Eggshells should be strong enough to protect the embryos during incubation, yet, fragile enough for late-stage embryos to hatch. Thus, mechanical strength of eggshell is an important factor for the reproductive biology of every oviparous amniote. Experimental compression or simulation studies have shown that eggshell thickness has a positive correlation with the strength of eggshells (*Ar et al., 1979*; *Hahn et al., 2017*; *Juang et al., 2017*; *López et al., 2021*), and this relationship may even provide a way to infer contact incubation in Palaeognathae (including extinct taxa; *Huynen et al., 2010*; *Yen et al., 2021*) as well as laying process of Palaeognathae (*Sellés et al., 2019*). However, *López et al., 2021* showed that microstructures, which was usually not considered in earlier studies such as finite element method, of avian eggshell can further contribute to the strength of the eggshells. We propose that testing the influence of different microstructures (e.g. rhea-style versus ostrich-style) in eggshells with similar thicknesses may provide further insights for the functional evolution of palaeognath eggshells (*Figure 14*). For example, *Hahn et al., 2017* showed that the average tensile failure stress of eggshell decreases with increasing egg size (and, typically, increased eggshell thickness) (but see also *Chiang et al., 2021* for elastic modulus). The high aspect ratio of thick palaeognath eggshell (*Figure 15*) may facilitate this relationship and compensate in fragility for the thick eggshells for late-stage embryos because think eggs are hard to break from inside.

The abundance of palaeognath eggshells in Cenozoic deposits makes it a biostratigraphically meaningful fossil (*Stidham, 2004*; *Harrison and Msuya, 2005*), but their microstructure and crystallography have rarely been studied. Palaeognath eggshells are widely distributed in Cenozoic deposits with palaeontological or archaeological significance across Africa, Asia, Europe, and Oceania. The eggshells have been conventionally differentiated into 'struthionid' and 'aepyornithid' types based on the shape of pore openings (*Sauer, 1972*). This simple criterion has been widely adopted in subsequent studies (*Sauer and Rothe, 1972*; *Stern et al., 1994*; *Harrison and Msuya, 2005*; *Donaire and López-Martínez, 2009*; *Patnaik et al., 2009*; *Wang et al., 2011*; *Pickford, 2014*; *Blinkhorn et al., 2015*; *Field, 2020*; *Mikhailov and Zelenkov, 2020*). However, (*Hirsch et al., 1997*, p. 363) stated that "The 'struthionid' and 'aepyornithid' pore system … should not be used solely in the identification and classification of eggshell". Furthermore, slit-like ( = 'aepyornithid') and circular pores ( = 'struthionid') coexist in some Neogene palaeognath eggshell fragments (*Bibi et al., 2006*; *Pickford, 2014*) and Quaternary moa eggshells (*Gill, 2007*). Potentially, the two different pores may represent just different parts of the egg, at least in some species (*Bibi et al., 2006*). Instead, we suggest that microstructural and crystallographic approaches presented in this study would provide a better basis for identifying and archiving poorly understood Cenozoic palaeognath eggshells.

Palaeoenvironmental information can be acquired from eggshells (*Stern et al., 1994*; *Angst et al., 2015*; *Montanari, 2018*; *Niespolo et al., 2020*; *Niespolo et al., 2021*; *Leuzinger et al., 2021*). Because Cenozoic palaeoenvironmental or geological events that might have influenced the evolution of Palaeognathae are comparatively well understood (*Mitchell et al., 2014*; *Claramunt and Cracraft, 2015*; *Grealy et al., 2017*; *Yonezawa et al., 2017*; *Crouch and Clarke, 2019*; *Figure 13*), further analytical investigation on Cenozoic palaeognath eggshells with proper geological and climatological contexts may shed light on the palaeoenvironmental settings of fossil localities and their effects in the evolution of Palaeognathae and its eggshells.

Zooarchaeology (or anthrozoology) is an additional serendipitous field that can be benefited by thorough understandings of palaeognath eggshells. Palaeognath eggs were not only important food resource for hunter-gatherers (*Oskam et al., 2011*; *Collins and Steele, 2017*; *Diehl et al., 2022*) but were also used for cultural purposes such as ornaments or storage containers (*Texier et al., 2010*; *Langley, 2018*; *Wilkins et al., 2021*; *Miller and Wang, 2022*), thereby, they are common in archaeological sites. Because chronological and palaeoenvironmental information inscribed in palaeognath eggshells in archaeological sites are available through isotopic analyses (*Sharp et al., 2019*; *Niespolo et al., 2020*; *Niespolo et al., 2021*), detailed microstructural information for those eggshells may provide more colourful implications (e.g. identification, harvest timing of egg, and biostratigraphy) about the interactions between early human, specific palaeognath avifauna, palaeoenvironments, and the precise age of palaeognath eggshell materials (e.g. *Harrison and Msuya, 2005*; *Loewy et al.,*

*2020*; *Niespolo et al., 2021*; *Douglass et al., 2021a*; *Douglass et al., 2021b*). For this, a solid understanding for microstructural evolution of modern palaeognath eggshells can be a helpful basis.

## Materials and methods

### Materials

Eggshells of all major clades of modern Palaeognathae were analysed (at least ten species including some that became extinct in Holocene): ostrich (*Struthio camelus*), rhea (*Rhea* sp.), emu (*Dromaius novaehollandiae*), cassowary (*Casuarius casuarius*), kiwi (*Apteryx mantelli*), elephant bird (Aepyornithidae), at least two species of moa (*Dinornis novaezealandiae* and either *Euryapteryx curtus* or *Pachyornis geranoides*; see Appendix 1; *Gill, 2010*; *Huynen et al., 2010*), and two species of tinamou (*Eudromia elegans* and *Nothoprocta perdicaria*). The materials represent the personal collection of YNL (ostrich, rhea, emu, elephant bird, and Chilean tinamou [*N. perdicaria*]); personal collection of MEH (cassowary); sourced from the Rainbow Springs Kiwi Sanctuary in Rotorua, New Zealand (kiwi); sourced from the Bronx Zoo, New York (elegant-crested tinamou [*E. elegans*]); and sourced from the Auckland War Memorial Museum in Auckland, New Zealand (moa). An eggshell of Paleocene palaeognath *Lithornis celetius* was analysed to acquire the data of fossil palaeognath eggshell. This material was excavated from the Fort Union Formation, Montana (*Weaver et al., 2022*), and its polarized light microscopic and scanning electron microscopic micrographs were presented in *Houde, 1988* and *Grellet-Tinner and Dyke, 2005*, respectively. The material (YPM 16961) was provided by Yale Peabody Museum of Natural History (New Haven, CT, USA).

Eggshells of five species of Neognathae, of which EBSD results were not available or insufficiently reported elsewhere, were analysed to provide information of non-palaeognath Neornithes (see *Choi et al., 2019*, table 1). The eggshells of three species were presented in the main text (*Figure 13*): common pheasant (Galliformes: *Phasianus colchicus*), northern goshawk (Accipitriformes: *Accipiter gentilis*), and European green woodpecker (Piciformes: *Picus viridis*). Japanese quail (Galliformes: *Coturnix japonica*) and common murre (Charadriiformes: *Uria aalge*) eggshells are shown in the Appendix 3. The common pheasant and Japanese quail eggshells were purchased from a local market; eggshells of northern goshawk were provided by a private collector; European green woodpecker eggs were provided by the Delaware Museum of Natural History (Wilmington, DE, USA) and common murre eggshells were provided by Erpur Hansen (South Iceland Nature Research Center).

Four Late Cretaceous non-avian maniraptoran dinosaur eggshells were analysed to provide broad overview of eggshell evolution (*Figures 13, 15 and 16*; Appendix 4). Three oospecies (parataxonomic classification of fossil eggshell) are presented in the main text: *Prismatoolithus levis, Elongatoolithus* oosp., and *Macroelongatoolithus xixiaensis* (or *M. carlylei* sensu *Simon et al., 2018*). *Prismatoolithus levis* is an ootaxon of *Troodon formosus* (Troodontidae; *Varricchio et al., 2002*) and the materials are from an egg that contains an embryo (MOR 246; *Horner and Weishampel, 1988*; *Varricchio et al., 2002*; *Choi et al., 2022*). *Elongatoolithus* (MPC-D 100/1047) and *Macroelongatoolithus* (SNUVP 201801) are oviraptorosaur eggshells (*Norell et al., 1994*; *Choi et al., 2019*; *Bi et al., 2021*; *Xing et al., 2022*) and *Macroelongatoolithus* was laid by a giant oviraptorosaur (*Pu et al., 2017*). We also presented EBSD image of *Triprismatoolithus stephensi* (ES 101; *Appendix 4—figure 4*; *Figure 1—figure supplement 1*) to present additional prismatic microstructure of the Late Cretaceous eggshell. The egg-layer of *T. stephensi* is unknown but suggested to be laid by a theropod dinosaur (*Jackson and Varricchio, 2010*; *Agnolin et al., 2012*). We further propose maniraptoran affinity of *T. stephensi* based on the existence of a SqZ, a diagnostic character of maniraptoran eggshells (*Choi et al., 2019*).

### EBSD

The methodology of EBSD analysis followed established protocols of *Moreno-Azanza et al., 2013* and *Choi et al., 2019* except for a newly adopted aspect ratio analysis. See Appendix 2 for details. The data were presented in inverse pole figure, Euler, grain boundary mappings, and misorientation distribution histograms. We had taken more than three maps (to assess the reproducibility of our observations) and misorientation distribution from a single eggshell, and results from the most well-prepared parts of the eggshell were presented.

In this study, aspect ratio mapping (*Koblischka-Veneva et al., 2010*) was introduced, which was successfully used to analyse the grain shape of brood parasitic and host eggshells (*López et al.,*

*2021*). In this method, a calcite grain is approximated as an ellipse. Based on the ratio of long to short axes of the ellipse, the grain is assigned to a colour level. This way, the aspect ratio of calcite grains can be quantitatively presented. We measured aspect ratio of all calcite grains in the maps. However, grains that are out of 50th percentile in area are presented in aspect ratio histograms. This step was necessary because smaller grains usually have a rounder shape and are quantitatively dominant compared to larger and more representative grains.

### Data analysis

All statistical analyses were performed in R 4.1.2 (*R Development Core Team, 2022*) on each of two distinct calibrated phylogenies for Palaeognathae, taken respectively from *Yonezawa et al., 2017* and *Kimball et al., 2019*. Log-transformed mean values were compiled for misorientation and aspect ratio, and used to perform ancestral state reconstructions on both phylogenetic trees for each trait – that is four distinct reconstructions (n=12 for all analyses). We assigned the three moa eggshell types to the species *Dinornis novaezealandiae*, *Euryapteryx curtus*, and *Pachyornis geranoides*, respectively (Appendix 1). Trees from *Yonezawa et al., 2017* and *Kimball et al., 2019* did not sample the three moa species in our dataset, but did sample their respective sister groups among moa (*Baker et al., 2005*; *Bunce et al., 2009*; *Huynen and Lambert, 2014*), allowing us to use their respective calibrations for each of them without altering the topology of either tree. Prior to each reconstruction, we estimated phylogenetic signal using Pagel's lambda (*Pagel, 1999*) in 'phytools' (*Revell, 2012*) to estimate how strongly the trait of interest follows a Brownian Motion model on the phylogeny of interest. In addition, we fitted different evolutionary models to the data and estimated their goodness of fit based on Akaike Information Criterion corrected for small sample sizes (AICc – *Burnham and Anderson, 2004*), using *fitContinuous* in 'geiger' (*Pennell et al., 2014a*) and *modSel.geiger* in 'windex' (*Arbuckle and Minter, 2015*), respectively. The fitted models (see e.g. *Mitchell et al., 2017*) include Brownian Motion (BM), Ornstein-Uhlenbeck (OU, single-optimum), Early Burst, Linear Trend, Lambda, and White Noise (i.e. a non-phylogenetic model). We did not test for more complex models (i.e. OU with multiple optima and/or selective regimes), as these are prone to high type I error for small sample sizes (*Cooper et al., 2016*).

For both trees, misorientation showed a high phylogenetic signal (Yonezawa: $\lambda$ =0.999; Kimball: $\lambda$ =0.987), with a BM model being selected as the best fit among tested evolutionary models. We thus performed ancestral reconstructions of misorientation following a maximum likelihood BM model using *contMap* in 'phytools' (*Revell, 2012*; *Revell, 2013*). Conversely, aspect ratio presented a low phylogenetic signal (Yonezawa: $\lambda$ =0.625; Kimball: $\lambda$ <0.001) and a White Noise model was always selected as the best fit. For the tree from *Yonezawa et al., 2017*, the BM model was selected as the second-to-best model with $\Delta$AICc <2, indicating it to be as good as the best model (*Burnham and Anderson, 2004*; *Richards, 2005*; *Symonds and Moussalli, 2011*); we thus also used *contMap* to reconstruct ancestral states of aspect ratio on this tree. For the tree from *Kimball et al., 2019*, however, the BM model was selected as the second-to-best model with $\Delta$AICc >2. This suggests that any optimization of aspect ratio on this tree would not reflect a true evolutionary pattern for our sample; we therefore did not perform this ancestral reconstruction.

## Acknowledgements

We thank Yong Park for extracting data; Matt Rayner, Ruby Moore, Vanessa Rhue, Daniel Brinkman, Brian Gill, Daniel Hanley, Erpur Hansen, Jungwoo Lee, and Jiwon Keum for sourcing eggshell materials. We thank Senior editor George Perry and Reviewer Albert G Sellés who provided helpful comments for the manuscript. Funding: SC was supported by the Basic Science Program through the National Research Foundation of Korea funded by Ministry of Education (2020R1A6A3A03038316) and the International Partnership Program of Chinese Academy of Sciences (132311KYSB20180016); MEH was supported by the Humboldt Foundation, Germany; LJL was supported by the Howard Hughes Medical Institute through the Science Education Program (GT10473); YNL was supported by the National Research Foundation of Korea (2019R1A2B5B02070240). The funders had no role in study design, data collection, and interpretation.

## Additional information

### Funding

| Funder | Grant reference number | Author |
| --- | --- | --- |
| Ministry of Education | 2020R1A6A3A03038316 | Seung Choi |
| Chinese Academy of Sciences | 132311KYSB20180016 | Seung Choi |
| Humboldt Foundation | | Mark E Hauber |
| Howard Hughes Medical Institute | GT10473 | Lucas J Legendre |
| National Research Foundation of Korea | 2019R1A2B5B02070240 | Yuong-Nam Lee |

The funders had no role in study design, data collection and interpretation, or the decision to submit the work for publication.

### Author contributions

Seung Choi, Conceptualization, Data curation, Funding acquisition, Investigation, Visualization, Methodology, Writing - original draft, Writing - review and editing; Mark E Hauber, Conceptualization, Resources, Supervision, Funding acquisition, Validation, Writing - review and editing; Lucas J Legendre, Data curation, Formal analysis, Funding acquisition, Visualization, Methodology, Writing - original draft, Writing - review and editing; Noe-Heon Kim, Investigation, Writing - review and editing; Yuong-Nam Lee, Resources, Supervision, Funding acquisition; David J Varricchio, Resources, Supervision, Funding acquisition, Validation, Project administration, Writing - review and editing

### Author ORCIDs

Seung Choi http://orcid.org/0000-0002-9013-2909
Mark E Hauber http://orcid.org/0000-0003-2014-4928
Lucas J Legendre http://orcid.org/0000-0003-1343-8725

### Decision letter and Author response

Decision letter https://doi.org/10.7554/eLife.81092.sa1
Author response https://doi.org/10.7554/eLife.81092.sa2

## Additional files

### Supplementary files

• MDAR checklist

### Data availability

All EBSD data generated or analysed during this study were uploaded in Dryad.

The following dataset was generated:

| Author(s) | Year | Dataset title | Dataset URL | Database and Identifier |
| --- | --- | --- | --- | --- |
| Choi S, Hauber ME, Legendre LJ, Kim N-H, Lee Y-N, Varricchio DJ | 2022 | Data from: Microstructural and crystallographic evolution of palaeognath (Aves) eggshells | https://doi.org/10.5061/dryad.dfn2z3550 | Dryad Digital Repository, 10.5061/dryad.dfn2z3550 |

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

# Appendix 1

## Moa eggshells used in this study and the number of species

Moa eggshells used in this study were collected from the North Cape, northern end of North Island, New Zealand (see *Gill, 2010*, Figure 1). The moa eggshells used in this study were given catalogue number LB8510 in *Gill, 2010* and we received thin (~0.90 mm), middle (~1.05 mm), and thick (~1.4 mm) eggshells from the Auckland War Memorial Museum. *Gill, 2010* provided a thickness histogram of North Cape moa eggshells and it showed a bimodal distribution (*Gill, 2010*, Figure 3). It has "a spread of numerous thin eggshell fragments (mode at 0.90–0.94 mm) and a second spread of rarer thicker fragments (mostly 1.2–1.7 mm thick)" (*Gill, 2010*, p. 117). The thicknesses of our three materials fall into thin and thick ranges of *Gill, 2010*.

In North Cape, majority of moa bone fossils are attributable to *Euryapteryx curtus* and *Pachyornis geranoides* (*Gill, 2022*), both of them are small- to medium-sized moa (*Gill, 2000*; *Bunce et al., 2009*). The presence of large moa *Dinornis novaezealandiae* (*Gill, 2000*; *Bunce et al., 2009*) was also reported from the North Cape based on body fossils (*Gill, 2010*). *Gill, 2022* provided key to the identification of North Island moa eggshells. According to the key, moa eggshells thinner than 1.0 mm with slit-shaped pore depression absent or short most likely belong to *P. geranoides*, eggshells range from 1.0 to 1.3 mm with slit-shaped pore depression most likely belong to *E. curtus*, and eggshells thicker than 1.3 mm with numerous and long slit-shaped pore-depression belong to *D. novaezealandiae*. According to these criteria, our thick eggshell unequivocally belongs to *D. novaezealandiae*. In the case of thin eggshell, *P. geranoides* is the most probable egg-layer because of the thickness (~0.90 mm) and near absence of slit-shaped pore depression. The middle eggshell probably belongs to *E. curtus* based on the thickness and pore structure. However, considering the variation of pore pattern and thickness within *Euryapteryx* eggshell (*Gill, 2022*, Figure 1) the thin and middle eggshells may belong to a single species. Therefore, we conclude that our moa eggshell materials derived from *D. novaezealandiae* and at least one smaller species (*E. curtus* or *P. geranoides*). The presence of three species in the North Cape inferred from body fossil record and haplotype data (*Gill, 2010*; *Huynen et al., 2010*) is consistent with palaeobiogeography of these taxa inferred from molecular phylogeny (*Bunce et al., 2009*).

It is noteworthy that *Oskam et al., 2011* criticized the use of eggshell thickness as a diagnosis of egg-laying moa species because intraspecific thickness variation among a certain species' eggshell is rather large so that it is hard to differentiate different types of moa eggshells. This criticism is reasonable enough as admitted by *Gill, 2022*, but the clear microstructural and crystallographic contrasts between the thin and thick moa eggshells used in this study (*Figures 9–11*) strongly support that our materials from the North Cape represent at least two different species of moa.

## Appendix 2

### Detailed methodology for EBSD analysis

The eggshells were embedded in epoxy resin with a drop of hardener and let them consolidate in room temperature for one day. After that, they were cut to expose radial sections using a circular blade. The rough exposed sections were lapped using 400-, 1000-, and 3000-grit aluminum compound by hand. Because complete polishing is crucial in EBSD analysis, the sections were polished by hand with 0.5 μm diamond paste for 20 min for each specimen. Finally, each specimen was polished with colloidal silica (0.06 μm) for 20 min using a turntable. The completed radial sections were coated with carbon. EBSD analyses were conducted using a FE-SEM (JEOL JSM-7100F) and its attached EBSD (Symmetry Detector; Oxford Instruments), housed in the School of Earth and Environmental Sciences, Seoul National University. The accelerating voltage of FE-SEM was 15.0 kV. EBSD analysis was performed in working distance 15.0 mm; 70 degrees tilting of the specimen. The Kikuchi lines were indexed using AZtec software (Oxford Instruments) and step size ranged from 0.25 μm (in case of tinamou eggshell) to 3 μm (in case of elephant bird eggshell), depending on the thickness of the eggshells.

The acquired mapping images were enhanced to correct wild spikes and unindexed pixels. A wild spike is an erroneous pixel, which is surrounded by correctly indexed pixels. The wild spikes were all eliminated. An unindexed pixel (or zero solution) is a failed indexing caused by no input of clear crystallographic data (=Kikuchi line) or several nearby grains information overlapped in a pixel, thus the AZtec software failed to read a proper signal. Following the method of *Choi et al., 2019*, we treated the unindexed pixel as the same signal of surrounding pixels when an unindexed pixel is surrounded by at least six consistent signals. We iteratively applied zero solution correction for three times.

The acquired EBSD data were presented in four different mappings (*Figures 1–12*; inverse pole figure [IPF], Euler, grain boundary [GB], and aspect ratio [AR]) and in misorientation histograms. In IPF mapping, *c*-axis orientation of calcite in the eggshell is presented using colour index. When the *c*-axis of the calcite grains lie perpendicular to the eggshell surface, the grain is coloured red. In contrast, when the *c*-axis of calcite grain lies parallel to the eggshell surface, it is coloured green or blue. In a Euler map, differently oriented calcite grains have different colours, thus it is easy to differentiate the individual calcite grains. In GB mapping, grain boundaries between the grains (i.e. misorientation) that are 5°–10°, 10°–20°, and >20° are marked with green, blue, and purples lines, respectively. The grain boundary information is used in constructing the misorientation histograms. In AR mapping, a calcite grain is approximated to an ellipse that has the most similar shape to the calcite grain. When the approximated ellipse is elongated, it is coloured green, yellow and even orange, but when an approximated ellipse is round, it is coloured blue. In making misorientation histograms, neighbour-pair and random-pair distributions were used. The former was acquired by selecting the adjacent pairs of grains and the angle between them are calculated, whilst the latter was calculated by using the randomly selected (hence, usually distant) grains. We selected 1000 randomly chosen grains in constructing the random-pair misorientation to be consistent with former studies, which presented misorientation histogram of fossil and modern eggshells (*Moreno-Azanza et al., 2013*; *Choi and Lee, 2019*; *Choi et al., 2019*; *Choi et al., 2020*; *Choi et al., 2022*).

## Appendix 3

### Detailed description for EBSD maps of neognath eggshells

Common pheasant (*Phasianus colchicus*) and Japanese quail (*Coturnix japonica*) (*Appendix 3—figures 1 and 2*): Both species belong to Galliformes and the microstructures and crystallography are similar enough to be described together. As in chicken and duck eggshells (*Choi et al., 2019*), the vertical alignment of *c*-axis is not as strong as that of palaeognath eggshells. The ML is wedge-like and the prismatic calcites extend to SqZ and EZ. The boundaries between the ML and SqZ, SqZ and EZ are gradual. As in ostrich-style palaeognath eggshells, a 'splaying' structure is very weak in SqZ. Low-angle GB are rare compared to palaeognath eggshells. ML and EZ have linear GB while SqZ has rugged GB. However, the difference in ruggedness between the layers is not prominent compared to that of palaeognath eggshells. These features are consistent with other galliform eggshells (chicken, *Choi et al., 2019*; megapode, *Grellet-Tinner et al., 2017*; turkey, *Nys et al., 2004*), but guineafowl eggshell has prominent rugged GB in SqZ as in rhea-style palaeognath eggshell (*Nys et al., 2004*).

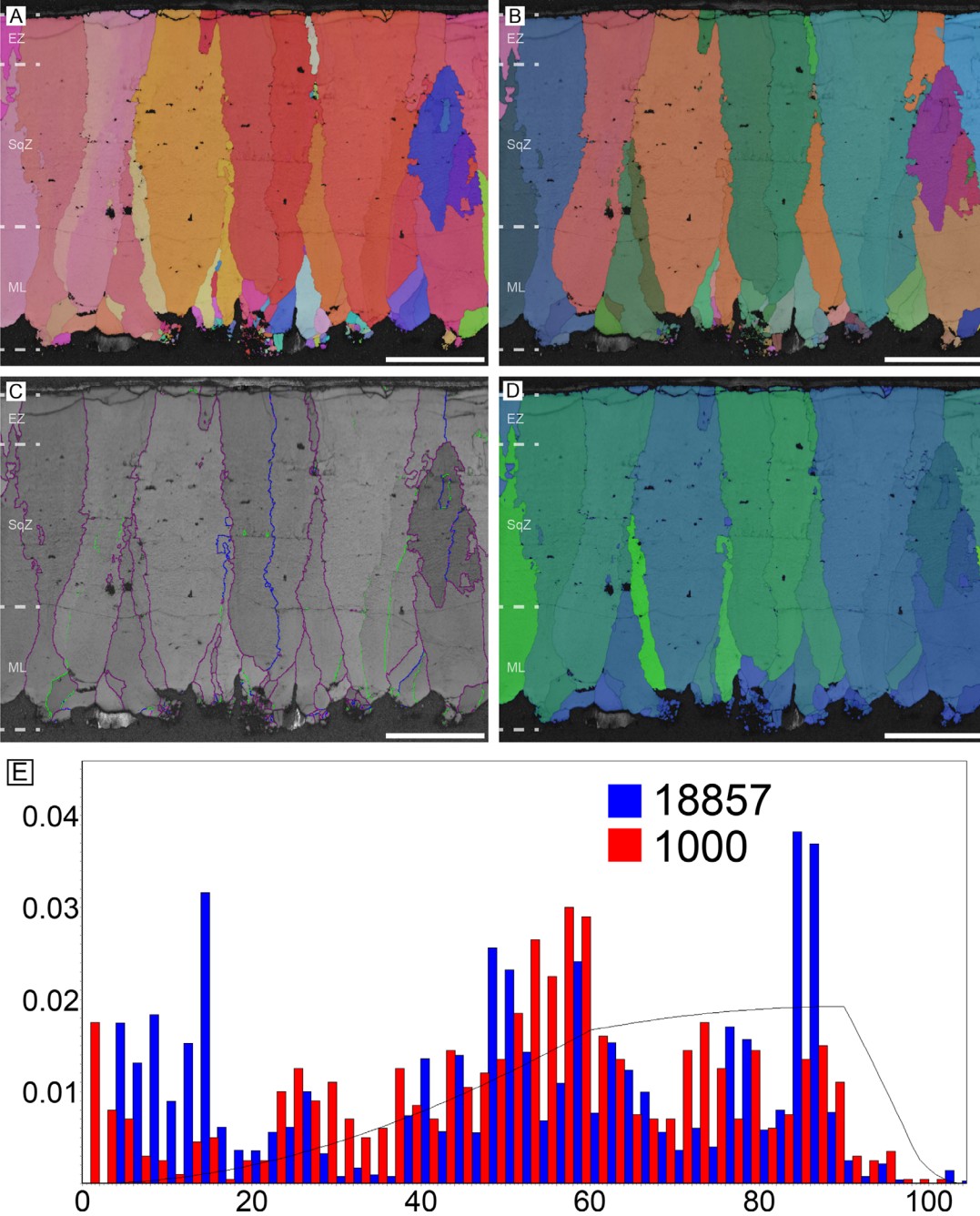

**Appendix 3—figure 1.** Common pheasant eggshell. Scale bars equal 100 μm.

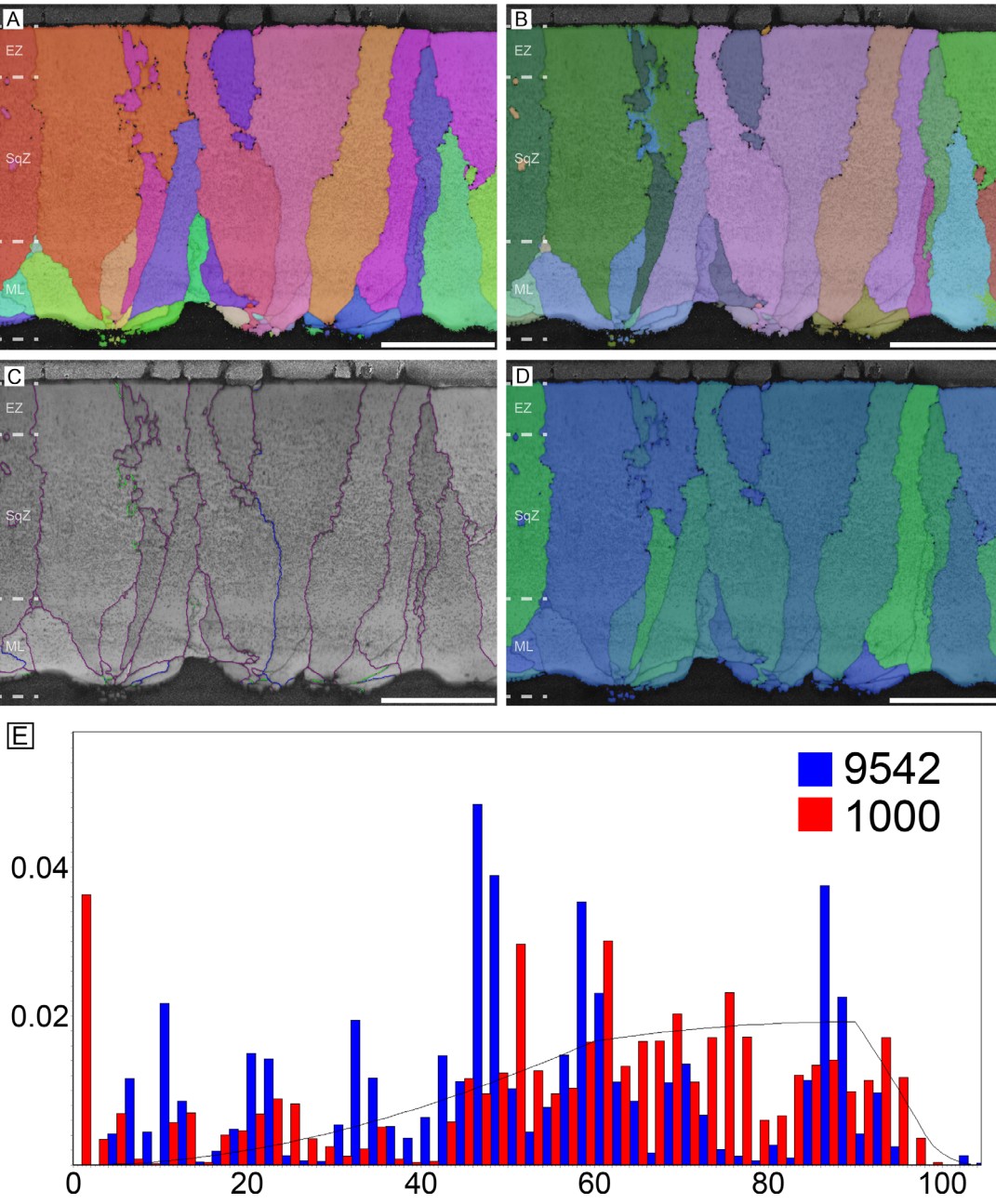

**Appendix 3—figure 2.** Japanese quail eggshell. Scale bars equal 50 µm.

Northern goshawk (*Accipiter gentilis*) (*Appendix 3—figure 3*): The goshawk eggshell has exotic microstructure. The ML is composed of wedge-like calcite. The boundary between the ML and SqZ is gradual. The SqZ seems to be thin while EZ is thick based on GB condition. In outer EZ, the calcite grains are characterized by angular porosities. This feature has not been observed in other avian eggshell and needs further investigation of other accipitriform eggshells (see also *Mikhailov, 1997a*). In addition, the porous EZ is overlain by another layer that is not indexed by EBSD. It means that this layer is not composed of calcite, but other biominerals. Preliminary energy-dispersive X-ray spectroscopy (EDS) analysis showed that Si is enriched in this layer. However, the origin and function of this layer is beyond the focus of current study. Low-angle GB are rarely seen. ML is composed of linear GB. The rugged GB is located near the inner part of the eggshell, while linear GB occupy nearly outer half of the calcite eggshell. This result may imply that the SqZ occupies only a small portion of the eggshell and EZ is dominant in the eggshell. Also, the angular porosities mentioned above may be confined to the EZ. However, in most avian eggshells, the vesicles are usually located

in SqZ, but in the northern goshawk eggshell, vesicles are widely distributed to the outer end of calcitic eggshell so identifying SqZ and EZ are equivocal. Along with the non-calcite biominerals mentioned earlier, testing this hypothesis is an unexplored issue in Accipitriformes biology, but we will not investigate this issue in this study. See also *Dalbeck and Cusack, 2006* for EBSD imaging for eggshell of another Accipitriformes, *Aquila chrysaetos*.

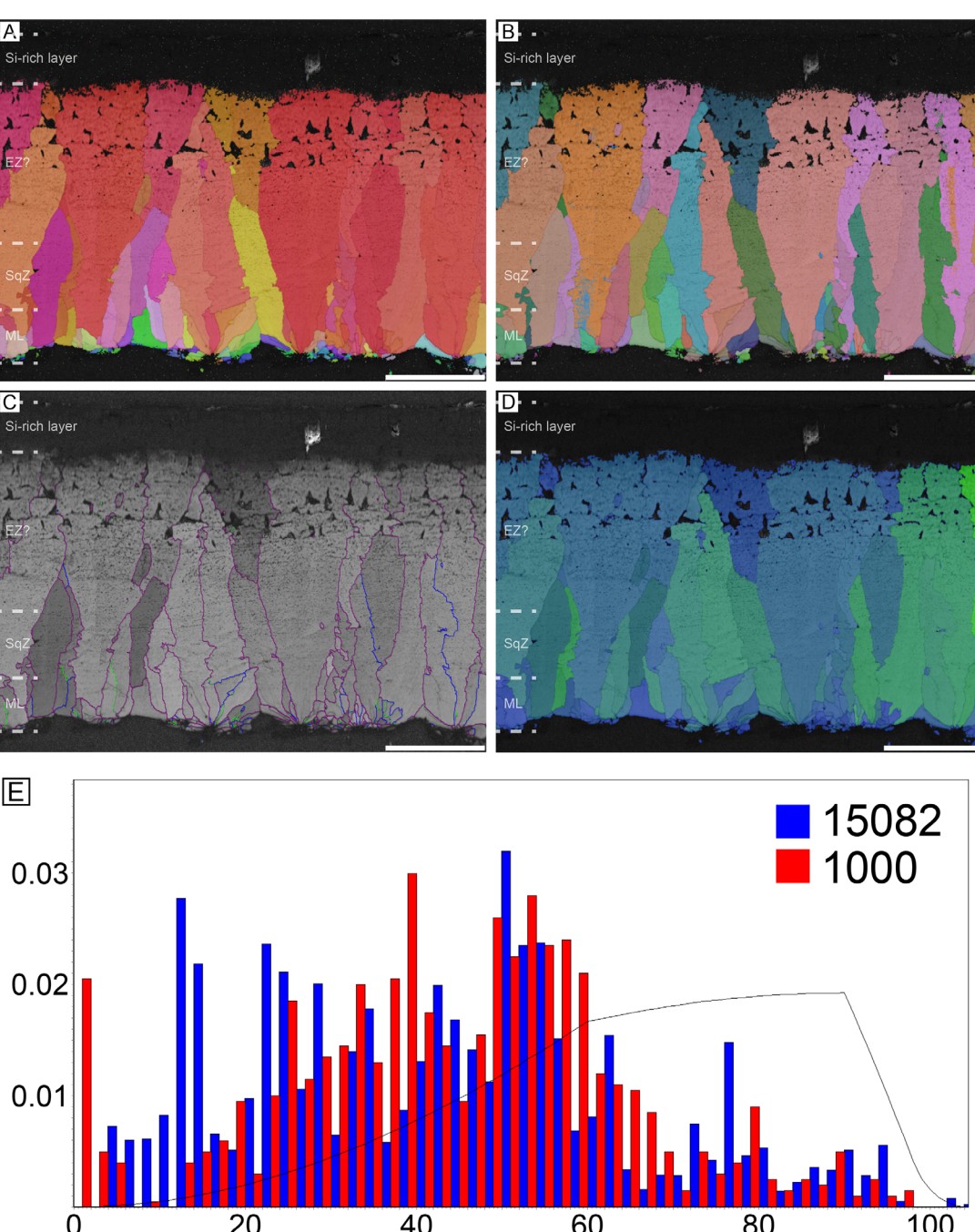

**Appendix 3—figure 3.** Northern goshawk eggshell. Scale bars equal 100 μm.

Common murre (*Uria aalge*) (*Appendix 3—figure 4*): Compared to the other neognath eggshells, the common murre eggshell has prominent vertical *c*-axis alignment, similar to that of palaeognath eggshells. The ML is wedge-like. Similar to palaeognath eggshell, the boundary between the ML and SqZ is easily identified due to the 'splaying' microstructure of SqZ. The calcite in the EZ is massive. Overall, common murre eggshell is remarkably similar to Guineafowl (Numididae, Galliformes) eggshell (*Nys et al., 2004*). Both eggshells are dissimilar to typical

'prismatic neognath morphotype' (*Mikhailov, 1997b*), but more like rhea-style palaeognath eggshell. Low-angle GB is usually confined to ML. In ML and EZ, GB are linear while in SqZ, GB are rugged.

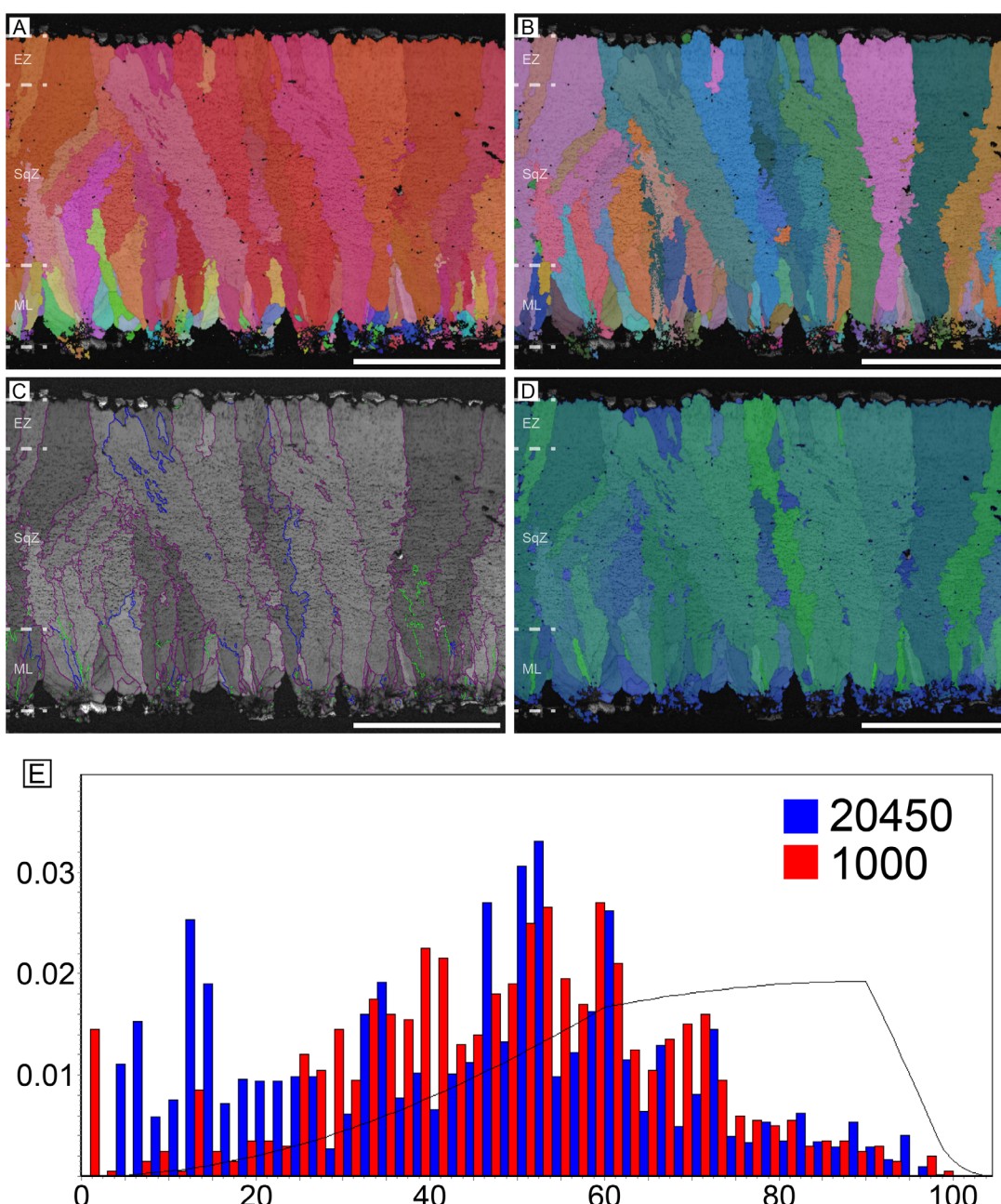

**Appendix 3—figure 4.** Common murre eggshell. Scale bars equal 250 µm.

European green woodpecker (*Picus viridis*) (*Appendix 3—figure 5*): The overall microstructure is strikingly similar to that of tinamou eggshells. It has relatively strong vertical *c*-axis alignment as palaeognath eggshells. The ML has needle-like structure although wedge-like structure is also observed. The boundary between the ML and SqZ is clear due to the contrasting microstructure. Grains are mostly irregular in SqZ. The EZ is composed of massive calcite grains. *Mikhailov, 1997a* already pointed out that eggshell of woodpecker (Neognathae) is similar to palaeognath eggshells. This observation can be further supported by the current study. Low-angle GB are concentrated on the ML. The GB configuration is similar to that of tinamou eggshell. The only minor difference is that the GB linearity in EZ is not as prominent as that of tinamou eggshells.

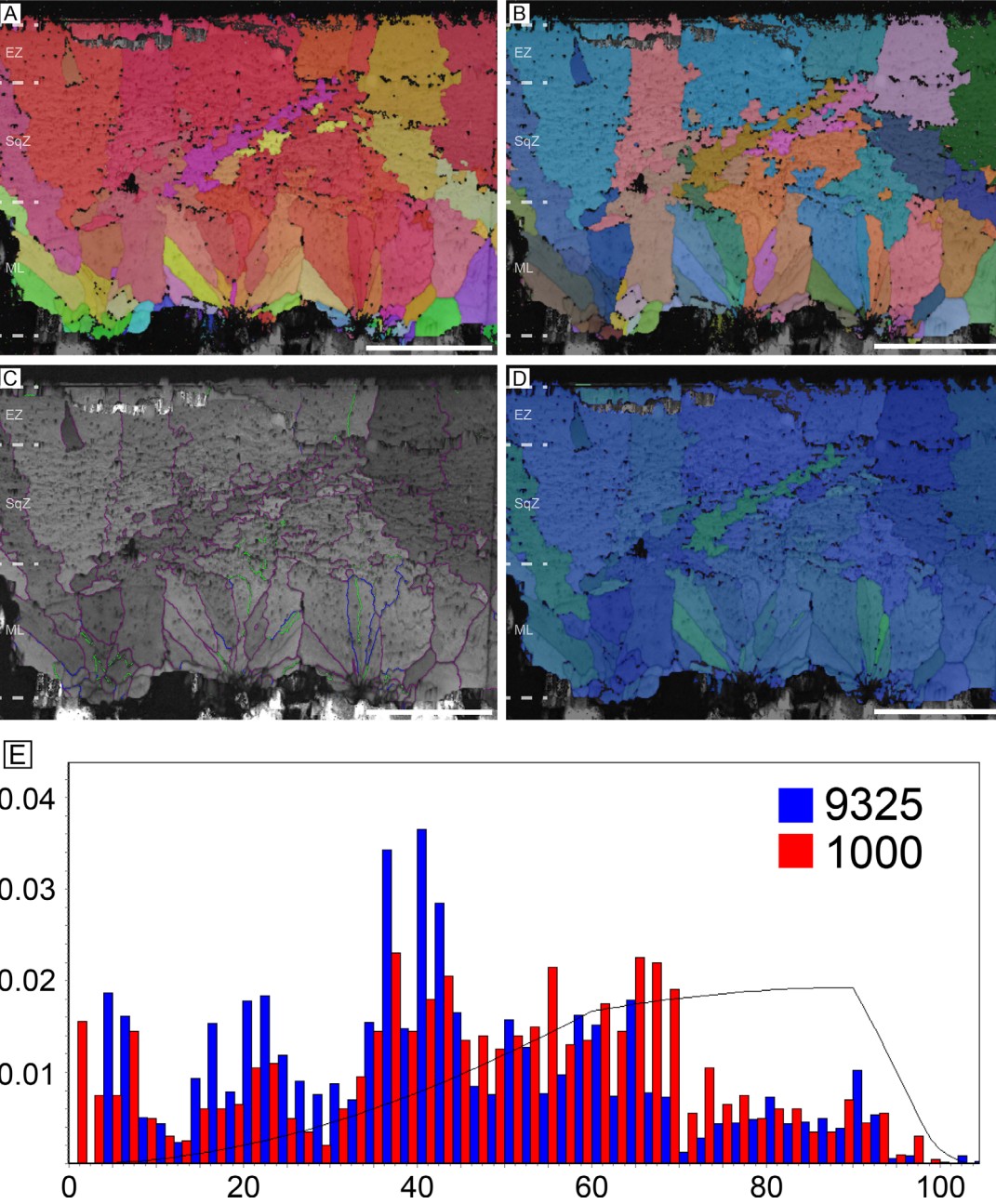

**Appendix 3—figure 5.** European green woodpecker eggshell. Scale bars equal 50 μm.

## Appendix 4

### Detailed description for EBSD maps of non-avian maniraptoran eggshells

The microstructural and crystallographic features of oviraptorosaur (*Elongatoolithus* oosp. and *Macroelongatoolithus xixiaensis* or *M. carlylei* sensu *Simon et al., 2018*; *Appendix 4—figures 1 and 2*) and troodontid eggshell (*Appendix 4—figure 3*) were fully described in *Choi et al., 2019*. Briefly, oviraptorosaur eggshells have rhea-style microstructure and crystallography except for the presence of needle-like ML and absence of EZ. This trait is more prominent for *Elongatoolithus*; *Macroelongatoolithus* is characterized by weakly developed, hidden prismatic structure (or cryptoprismatic structure sensu *Jin et al., 2007*). Oviraptorosaur eggshells have widespread low-angle GB, not confined to certain part of the eggshell. ML has linear GB, but in SqZ, GB becomes highly rugged. They do not have EZ that is characterized by linear GB. In contrast, troodontid eggshell (*Prismatoolithus levis*; *Zelenitsky and Hills, 1996*; *Varricchio et al., 2002*; *Varricchio and Jackson, 2004*; *Funston and Currie, 2018*) has well-developed prismatic shell units. In addition, *P. levis* has EZ (*Varricchio and Jackson, 2004*), a feature that is usually found in avian eggshell. Although the existence of EZ is still challenged (e.g. *Mikhailov, 2014*), general consensus is that EZ exists in *P. levis* (*Zelenitsky and Therrien, 2008*; *Choi et al., 2019*). Troodontid eggshell is characterized by linear GB in ML and EZ, but SqZ has very weakly developed rugged GB (*Figure 1—figure supplement 1*).

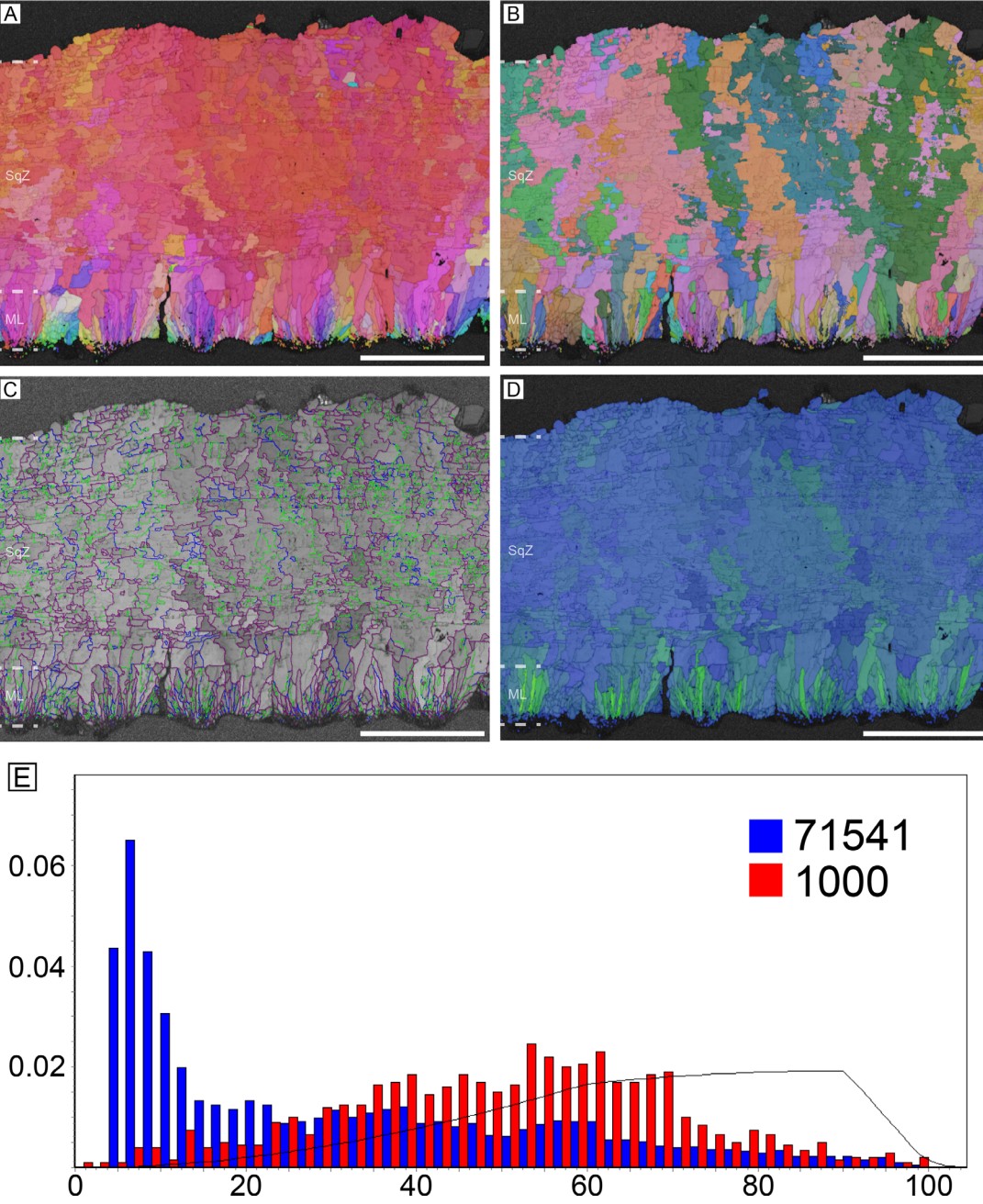

**Appendix 4—figure 1.** *Elongatoolithus* oosp. (see also *Choi et al., 2019*). Scale bars equal 250 μm.

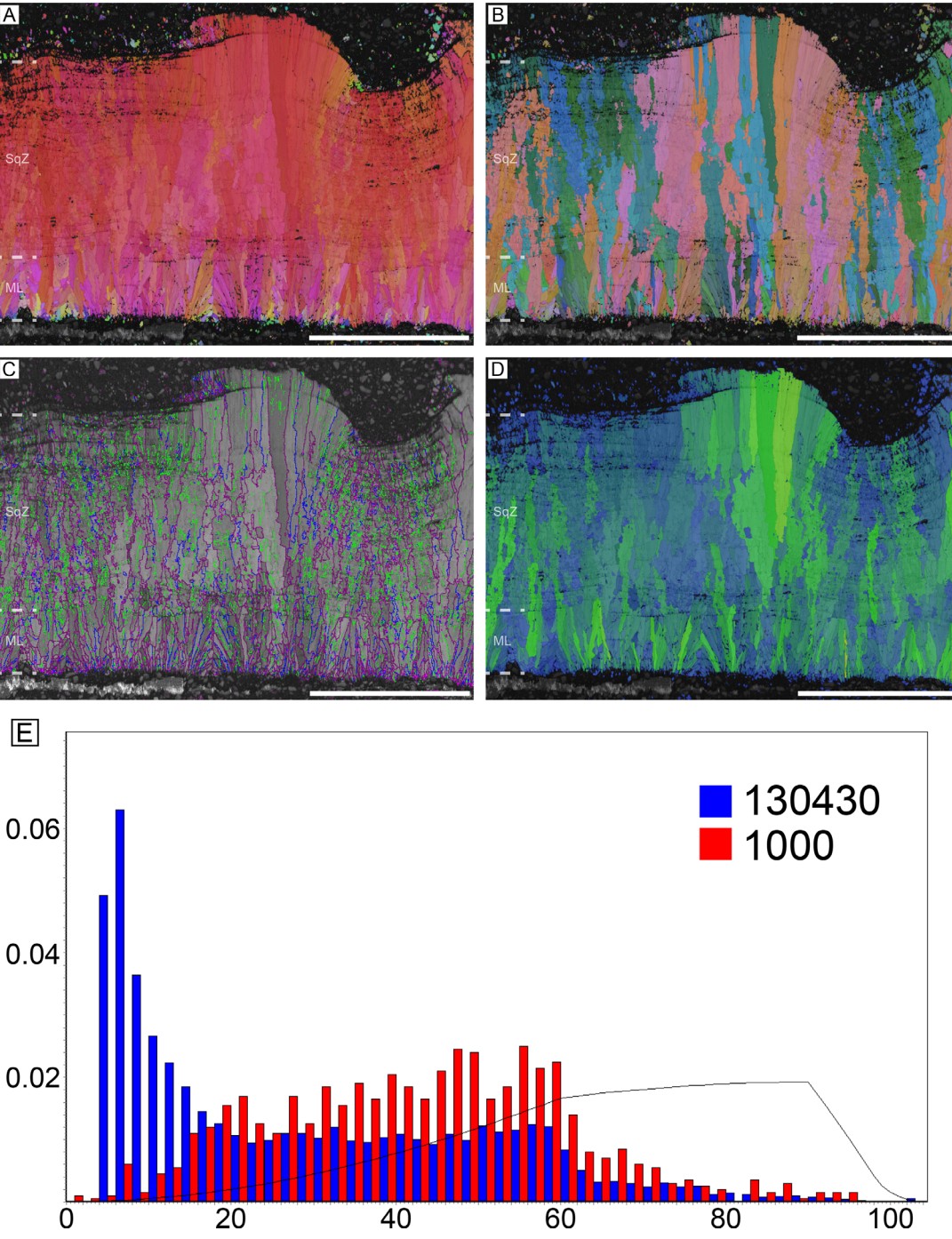

**Appendix 4—figure 2.** *Macroelongatoolithus xixiaensis* (or *M. carlylei*) (see also **Huh et al., 2014**; **Choi et al., 2019**). Scale bars equal 1000 μm.

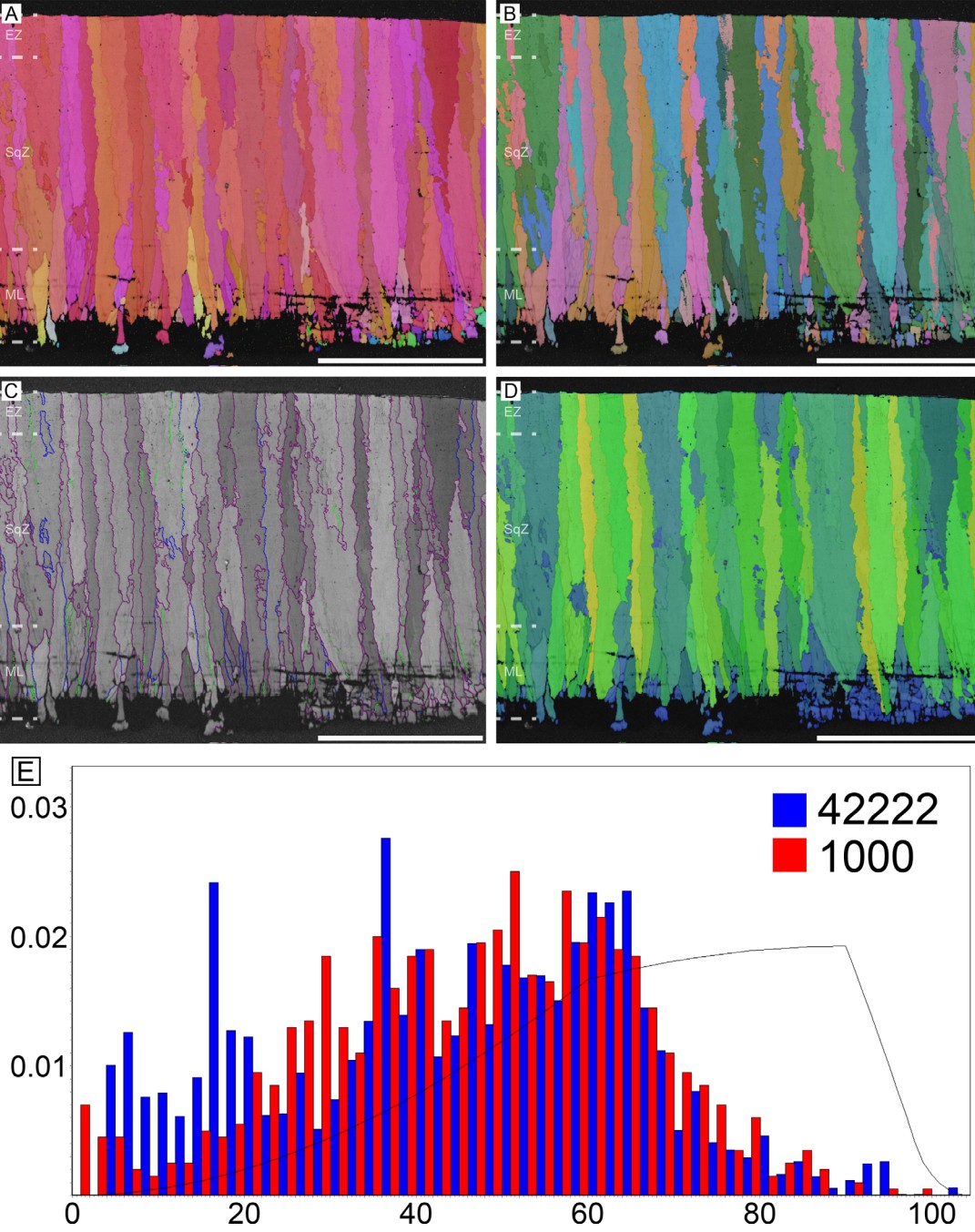

**Appendix 4—figure 3.** *Prismatoolithus levis* (see also ***Varricchio et al., 2002***; ***Varricchio and Jackson, 2004***). Scale bars equal 500 μm.

*Triprismatoolithus stephensi* (***Appendix 4—figure 4***): The overall microstructure is similar to that of *P. levis* in that prismatic shell units occupy the whole thickness of eggshell. The ML is wedge-like, and the boundary between the ML and SqZ is gradual. The existence of EZ can be identified by grain boundary condition and polarized light microscopic features (***Jackson and Varricchio, 2010***; ***Varricchio and Jackson, 2016***). The overall GB configuration is similar to that of *P. levis*. Low-angle GB are rare. In ML and EZ, grain boundaries are linear but in SqZ, grain boundaries are rugged. However, the difference in ruggedness of SqZ and EZ is clearer compared to that of *P. levis*.

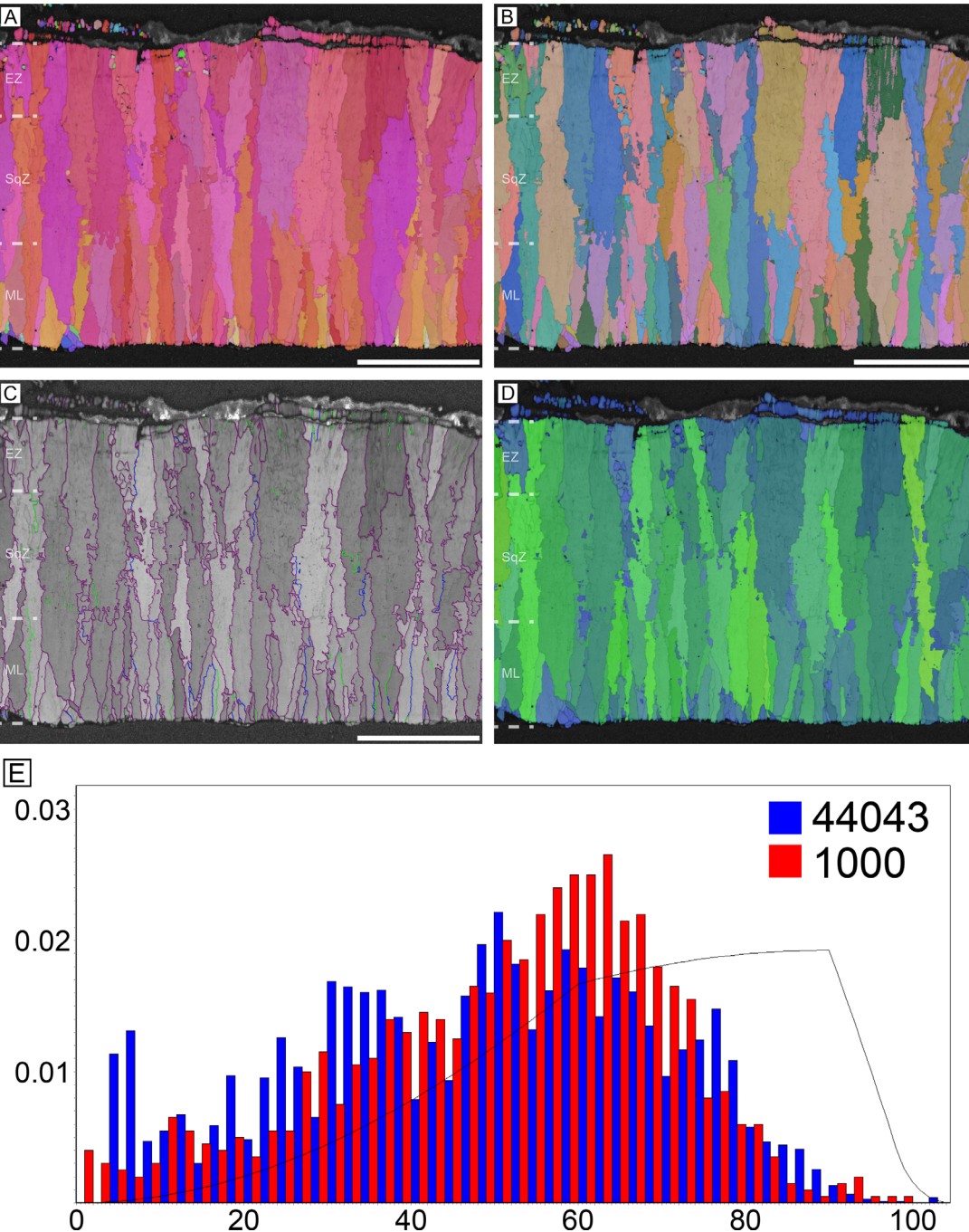

**Appendix 4—figure 4.** *Triprismatoolithus stephensi* (see also *Jackson and Varricchio, 2010*; *Yang et al., 2018*). Scale bars equal 250 µm.

## Appendix 5

### Further information for aspect ratio

Because it is known that *Prismatoolithus levis* is characterized by very narrow shell units (*Zelenitsky and Hills, 1996*; *Varricchio et al., 2002*; *Varricchio and Jackson, 2004*), whereas the shell unit structure of oviraptorosaur *Macroelongatoolithus* (*Pu et al., 2017*) is rather controversial (*Grellet-Tinner et al., 2006*; *Jin et al., 2007*), we analysed AR of oviraptorosaur (*Elongatoolithus* and *Macroelongatoolithus*) and troodontid eggshells (*P. levis*). Intriguingly, *P. levis* shows very high AR following the AR of ostrich eggshell (*Figure 15*). Both *P. levis* and modern ostrich eggshell have very narrow and tall shell units and weakly developed 'splaying' microstructure. *Elongatoolithus* does not have high AR whereas *Macroelongatoolithus* shows higher AR. It strongly supports the view that the SqZ of *Macroelongatoolithus* can be best described by the term 'cryptoprismatic' (*Jin et al., 2007*) but 'aprismatic' may be still a reasonable term for *Elongatoolithus* (*Grellet-Tinner and Chiappe, 2004*).

It was proposed that character states of ML, either needle-like or wedge-like, can be diagnosed by richness of low-angle GB (*Choi et al., 2019*). However, it can be better defined by the combination of AR and GB mappings of the ML (*Figure 7—figure supplement 1*). We suggest that density of calcite grains within a ML would be the most objective criterion. When the calcite grains of ML are confined to ML, the AR mapping provides most straightforward results: needle-like ML is characterized by the presence of high AR colour (e.g. *Reticuloolithus*, *Elongatoolithus*, kiwi, and tinamou eggshells). However, when the calcite grains of ML are not limited to ML, even wedge-like calcite grains show high AR colour (e.g. ostrich and thick moa eggshells). In this case, a wedge-like ML is composed of a small number of calcite grains, thus one can count the number of grains in a single ML. It will provide a way to circumvent the limitation of AR mapping in diagnosing wedge-like versus needle-like ML in eggshells. In sum, identifying needle- or wedge-like ML can be quantitatively done by counting calcite grains in ML in all cases; when the calcite grains are confined to ML and not extended to SqZ, AR mapping will provide the simplest visual representation. Detailed elaboration of this approach will be presented in a future research.

## Appendix 6

### Alternative interpretation

The alternative interpretations presented here are based on the phylogeny of *Cloutier et al., 2019* and *Sackton et al., 2019*; *Appendix 6—figures 1–3*. Only those differences from the main text are discussed herein: (i) the tinamou and moa are nested in a more inclusive position of the tree and rhea is assigned to a less inclusive position. Still, three tinamou-style and two ostrich-style microstructures might have been derived from rhea-style microstructure. (ii) Unfortunately, the rate of evolutionary change cannot be inferred in this interpretation because speciation timeline estimates have not been made in *Cloutier et al., 2019* nor *Sackton et al., 2019*.

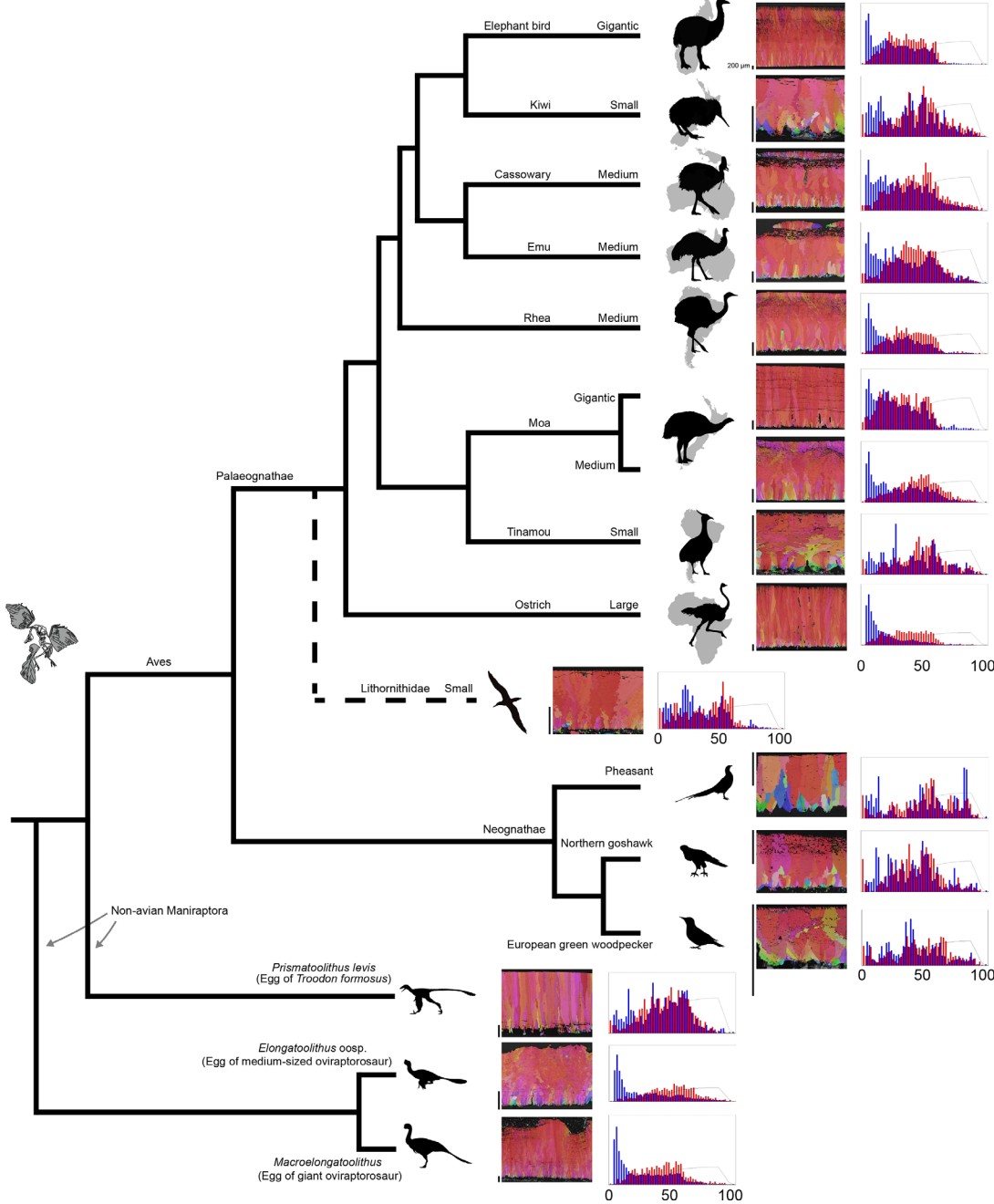

**Appendix 6—figure 1.** Alternative interpretation of phylogeny of Palaeognathae with IPF mapping and MD. After *Cloutier et al., 2019* and *Sackton et al., 2019*.

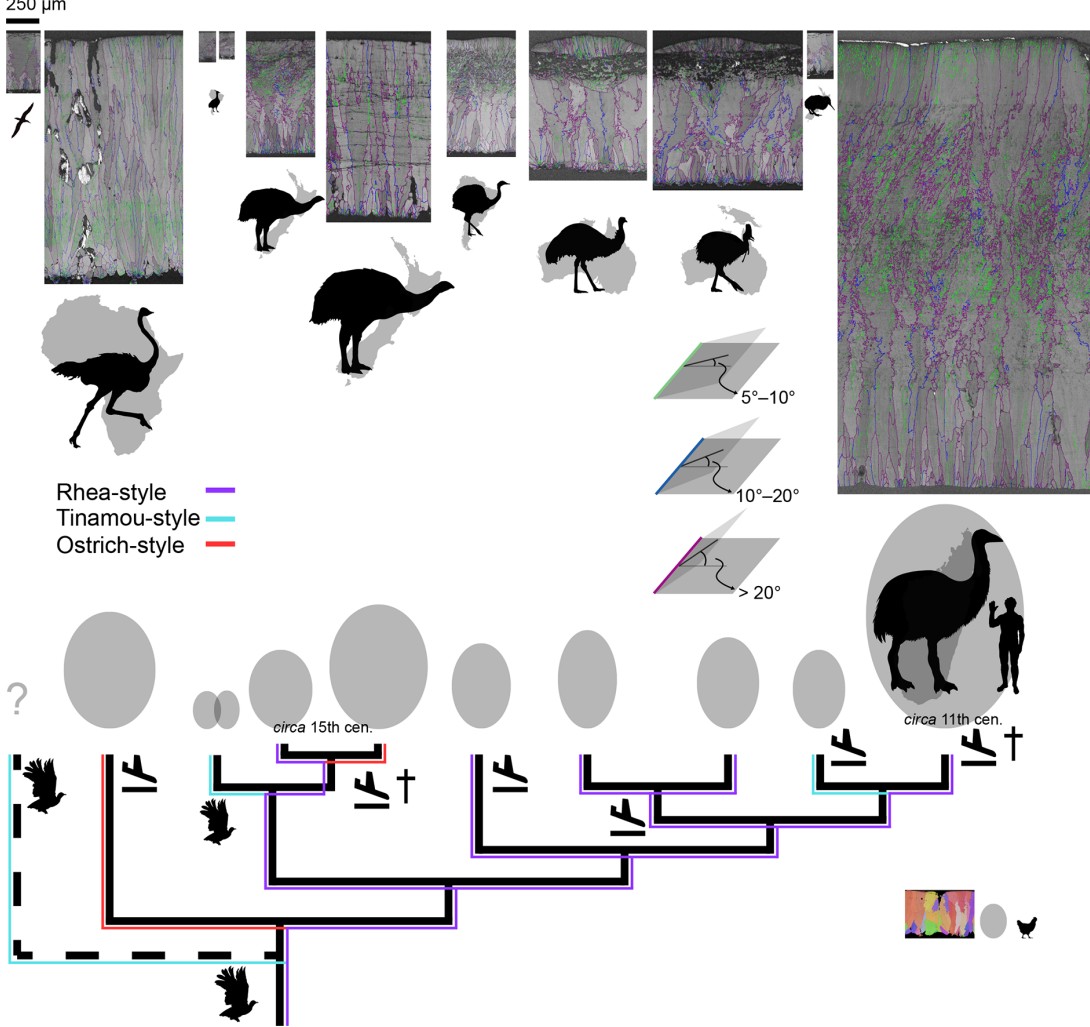

**Appendix 6—figure 2.** Alternative interpretation of phylogeny of Palaeognathae with GB mapping, egg size, and thickness of eggshells. After *Cloutier et al., 2019* and *Sackton et al., 2019*.

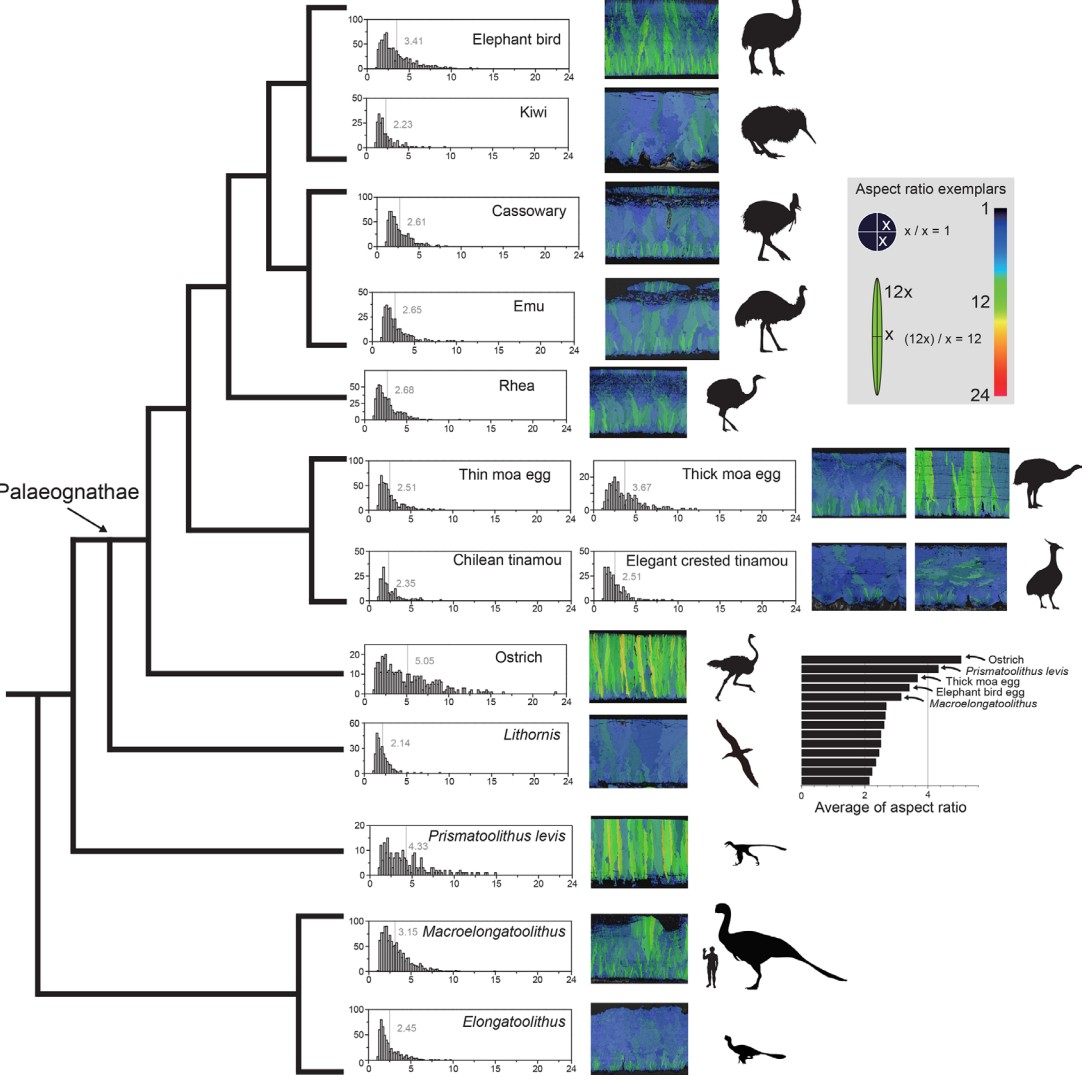

**Appendix 6—figure 3.** Alternative interpretation of phylogeny of Palaeognathae with AR mapping. After *Cloutier et al., 2019* and *Sackton et al., 2019*.

