## [Editor Report]

This fundamental study represents a significant advance in our understanding of the complex evolutionary history of the eggshell features in one of the main living bird lineages, Palaeognathae, with compelling and thoughtfully presented results. The work will be of interest to many biologists, paleontologists, and archaeologists.

---

## [Decision Letter]

**Decision letter after peer review:**

Thank you for submitting your article "Microstructural and crystallographic evolution of palaeognath (Aves) eggshells" for consideration by *eLife*. Your article has been reviewed by 1 peer reviewers, and the evaluation has been overseen by George Perry as the Senior Editor. The following individual involved in review of your submission has agreed to reveal their identity: Albert Sellés (Reviewer #1).

Essential revisions:

Please address the reviewer's comment regarding Triprismatoolithu stephensis, which is reported as analyzed by the authors, but then not included in any of the phylogenetic analyses or associated figures. Please explain and either include the specimen in the analyses or consider excluding it from the manuscript, accordingly.

*Reviewer #1 (Recommendations for the authors):*

I have no specific recommendation for the present study. As I already said, the study is well designed, the results are solid and well supported by the right methodology, and the discussion and conclusions are in accordance to the resulted data.

However, I would like to suggest that, for future studies, it would be interesting to analyse plausible palaeognathid eggshells from the early Paleogene, such as Incognithoolithus, Microolithus, and Metoolithus from the Eocene of North America (Kohring and Hirsch, 1996; Hirsch et al., 1997; Jackson and Varrichio, 2013), an include them in the scenario presented by Choi and co-authors.

In my humble opinion, they could help to elucidate the early acquisition of some morphologic and crystallographic features present in "modern" palaeognathes, or support some of the homoplastic hypothesis presented in the present work.

For all the above, I believe that the work can be published in its current state.

I only need to congratulate the authors for their work.

---

## [Author Response]

Essential revisions:Please address the reviewer's comment regarding Triprismatoolithu stephensis, which is reported as analyzed by the authors, but then not included in any of the phylogenetic analyses or associated figures. Please explain and either include the specimen in the analyses or consider excluding it from the manuscript, accordingly.

Because *Triprismatoolithus* has a role in Figure 1—figure supplement 1, which is important for the understanding of microstructure of ostrich and thick moa eggshells, we prefer maintaining *Triprismatoolithus*.

Reviewer #1 (Recommendations for the authors):I have no specific recommendation for the present study. As I already said, the study is well designed, the results are solid and well supported by the right methodology, and the discussion and conclusions are in accordance to the resulted data.However, I would like to suggest that, for future studies, it would be interesting to analyse plausible palaeognathid eggshells from the early Paleogene, such as Incognithoolithus, Microolithus, and Metoolithus from the Eocene of North America (Kohring and Hirsch, 1996; Hirsch et al., 1997; Jackson and Varrichio, 2013), an include them in the scenario presented by Choi and co-authors.In my humble opinion, they could help to elucidate the early acquisition of some morphologic and crystallographic features present in "modern" palaeognathes, or support some of the homoplastic hypothesis presented in the present work.For all the above, I believe that the work can be published in its current state.I only need to congratulate the authors for their work.

Thank you very much for sharing your interesting opinion, which will definitely be a very good follow-up study. As Reviewer 1 correctly points out, there are many probable palaeognath eggshells in the fossil record, and the ones from the early Paleogene will be useful to trace the evolution of palaeognath eggshells.

We would like to share our further opinions that may supplement the great research design of Reviewer 1:

Firstly, investigating the rhea-style eggshell microstructures among modern neognath eggshells is necessary for a better understanding. As we mentioned in Lines 350–356, some neognath eggshells have either ‘rhea-style’ or ‘tinamou-style’ microstructure and this fact had been also pointed out by Mikhailov (1997a). Currently, we (the authors) consider that at least some ‘rhea-style’ morphotype of NEOGNATH eggshells may be homologous to ‘rhea-style’ morphotype of palaeognath eggshell. IF this assumption is correct, when ‘rhea-style’ eggshells are found from the Paleogene deposits, a researcher may not be able to make a solid conclusion whether what he/she found are ‘rhea-style’ eggshells of PALAEOGNATHAE or ‘rhea-style’ eggshells of NEOGNATHAE. We consider that only if this obstacle is satisfactorily eliminated, then the design outlined by Reviewer 1 will be an unquestionable one. We palaeontologists should still pay more attention to modern avian eggshells.

Secondly, we would like to emphasize that southwestern Europe (especially Spain, specifically Catalonia) is a very promising region for the follow-up study. On the one hand, the ootaxon *Sankofa pyrenaica* and *Pseudogeckoolithus* (Lines 656–660), both have remarkable ‘rhea-style’ microstructure, are found from Catalonia. The microstructure of *Pseudogeckoolithus* has been analyzed with EBSD (Choi et al. 2020, *Papers in Palaeontology*), but the microstructure of *Sankofa* (López-Martínez and Vicens 2012, *Palaeontology*) has not. In our opinion, with the EBSD images of *Sankofa*, one can gather both microstructural and crystallographic information, and it will be very useful to comparative information for palaeognath data. If there are additional similarity other than microstructural characters, we believe that *Sankofa* may become one of the strongest candidates of potential palaeognath eggshells in the Cretaceous fossil record. On the other hand, *Ornitholithus* (Donaire and López-Martínez 2009, *Palaeo3*) is another ootaxon from Catalonia, and Figures 3 and 7 in Donaire and López-Martínez (2009) made us believe that this ootaxon is most probably palaeognath eggshell (IF above mentioned obstacle is clearly addressed) and its microstructure (Figure 3) may show the transition from ‘rhea-style’ to ‘ostrich-style’ (i.e. the ‘purple to red’ transition shown in our Figure 14). With the detailed understanding for the Catalonian ootaxon *Ornitholithus*, future researchers may be able to understand the evolution of palaeognath eggshells more vividly.

Thank you again for sharing interesting opinion with us, and we are happy to collaborate or share our data with Reviewer 1 if you are interested in the potential follow-up studies.